# DIFFERENTIALLY PRIVATE FEDERATED LEARNING WITH TIME-ADAPTIVE PRIVACY SPENDING

**Shahrzad Kiani**[1]***Nupur Kulkarni**[2], **Adam Dziedzic**[2], **Stark Draper**[1], **& Franziska Boenisch**[2]
[1] Department of Electrical and Computer Engineering, University of Toronto
[2] CISPA Helmholtz Center for Information Security

## ABSTRACT

Federated learning (FL) with differential privacy (DP) provides a framework for collaborative machine learning, enabling clients to train a shared model while adhering to strict privacy constraints. The framework allows each client to have an individual privacy guarantee, e.g., by adding different amounts of noise to each client's model updates. One underlying assumption is that all clients spend their privacy budgets uniformly over time (learning rounds). However, it has been shown in the literature that learning in early rounds typically focuses on more coarse-grained features that can be learned at lower signal-to-noise ratios while later rounds learn fine-grained features that benefit from higher signal-to-noise ratios. Building on this intuition, we propose a *time-adaptive* DP-FL framework that expends the privacy budget non-uniformly across both time and clients. Our framework enables each client to save privacy budget in early rounds so as to be able to spend more in later rounds when additional accuracy is beneficial in learning more fine-grained features. We theoretically prove utility improvements in the case that clients with stricter privacy budgets spend budgets unevenly across rounds, compared to clients with more relaxed budgets, who have sufficient budgets to distribute their spend more evenly. Our practical experiments on standard benchmark datasets support our theoretical results and show that, in practice, our algorithms improve the privacy-utility trade-offs compared to baseline schemes.

## 1 INTRODUCTION

With machine learning (ML) relying increasingly on users' sensitive data, the development of utility-driven frameworks that also adhere to users' individual constraints, including privacy preferences, has become a priority. When federated learning (FL) (McMahan et al., 2017) was first introduced, it was perceived as a privacy-preserving distributed learning framework that allows users (also called *clients*) to keep their data local and solely exchanging model updates with the server who can aggregate the updates and apply them to the global model for training.

However, it has been shown that data can be leaked through the model gradients (Zhu et al., 2019; Geiping et al., 2020; Boenisch et al., 2023). Hence, FL was extended to incorporate formal privacy guarantees (McMahan et al., 2017; Geyer et al., 2017; Wei et al., 2020; Hu et al., 2023; Ramaswamy et al., 2020) via the mathematical framework of differential privacy (DP) (Dwork, 2006). In DP-FL frameworks, one common approach is to protect the entire dataset of each client ("client-level DP") by clipping local model updates and adding noise before releasing them in each training round (Truex et al., 2019; 2020). This ensures that an adversary who has access to the aggregated perturbed updates of a subset of clients cannot confidently infer whether or not any particular client has participated in the given training round. However, although the DP-FL framework ensures privacy, it degrades model utility by introducing errors into the model updates. Thus, careful calibration of the perturbations is necessary to balance privacy and utility.

There are several extensions of the DP-FL framework in the literature that aim to improve privacy-utility tradeoffs by reducing the effect of perturbation while adhering to the privacy budgets of

---

*correspondence to shahrzad.kianidehkordi@mail.utoronto.ca. Part of the work was done while Shahrzad Kiani visited CISPA.

clients. Typically, these prior works (Pichapati et al., 2019; Yang et al., 2021; Shen et al., 2023; Yang et al., 2023; McMahan et al., 2017) consider the inherent heterogeneity in FL—both in data and privacy—to make perturbation more efficient. Yet, they either assume that clients' privacy budgets should be exhausted *uniformly over time* (Boenisch et al., 2024), or rely on strong assumptions regarding access to public data (Li et al., 2022) or negligible privacy loss when adjusting privacy parameters in a time-adaptive manner based on data (Pichapati et al., 2019). As we will see, judiciously expending the budget non-uniformly over time and in a privacy-preserved manner can yield an improved privacy-utility tradeoff. Hence, a gap exists in the literature.

In this work, we reduce this gap by proposing a novel DP-FL framework with a *data-independent time-adaptive* privacy spending method. In our framework, clients can spend their privacy budget *non-uniformly across time* (training rounds). This means that clients intentionally allocate less of their privacy budgets in the early rounds to save them for later rounds. We term these rounds as "saving". Clients then transit to "spending" rounds, wherein they uniformly allocate their remaining budget across spending rounds. The decisions about when each client transits from saving to spending and how much they save in each round are made solely based on clients' privacy budgets, and not their local data. Therefore, we can schedule spending before the start of training, making it free of privacy loss. We account for each client's privacy spending in each round, formulating privacy bounds as a function of clients' decisions and budgets.

Two observations that motivate the potential of our framework to improve the privacy-utility tradeoff are as follows. First, by preserving the privacy budget in early rounds and incrementally spending later, we are able to adjust the signal-to-noise (SNR) ratio to be uneven across training rounds, with noise shifting from later rounds to earlier ones. This enables coarse-grained features, which are typically learned in the early rounds and are more tolerant to noise, still to be learned effectively. Furthermore, the fine-grained features, typically learned in later rounds (Dziedzic et al., 2019; Raghu et al., 2017; Shwartz-Ziv & Tishby, 2017), can be learned in a beneficial higher-SNR setting. Secondly, in practical scenarios, we note that clients are likely to have different privacy budgets. We show theoretically that clients with stricter privacy budgets benefit from expending their privacy budgets more unevenly than those with relaxed (larger) budgets. Intuitively, this allows less-privacy-sensitive clients, who have often sufficient budgets, to contribute to the learning both of coarse-grained features in early rounds and of fine-grained features in later rounds. On the other hand, more-privacy-constrained clients can preserve their budgets and helpfully contribute more to the learning of fine-grained features.

In summary, we make the following contributions:

- As part of our framework design (detailed in Sec. 3), we introduce a novel privacy spending method, namely "spend-as-you-go", where clients spend their privacy budgets incrementally over time, instead of spending the privacy budget uniformly across time, as in traditional DP-FL approaches.
- Our theoretical analysis (detailed in Sec. 4) provides privacy accounting for the incremental spending pattern in our method. Additionally, we show theoretically, that if clients that use stricter privacy parameters, such as lower clipping norms, save a larger portion of their privacy budget during saving rounds, and can spend more in spending rounds, then, in expectation, we can reduce the clipping bias (Das et al., 2023).
- Based on these theoretical insights, in Sec. 5 we experimentally benchmark our framework against the baselines and show that the global test accuracy achieved by our method surpasses that of the baselines for the FMNIST (Xiao et al., 2017), MNIST (Deng, 2012), Adult Income (Becker & Kohavi, 1996), and CIFAR10 datasets (Krizhevsky et al., 2009).

## 2 BACKGROUND AND RELATED WORK

**Federated Learning.** We consider a typical FL system with $N$ clients and a central server. Each client $n \in [N]$ has its own data distribution $P_n$ on $\mathcal{X} \times \mathcal{Y}$, where $\mathcal{X} \subseteq \mathbb{R}^u$ denotes the feature space and $\mathcal{Y} \subseteq \mathbb{R}$ denotes the label space. Let $\mathcal{L}_n : \mathbb{R}^v \times \mathcal{X} \times \mathcal{Y} \to \mathbb{R}$ denote client $n$'s loss function which maps a model parameter $\theta \in \mathbb{R}^v$ and a data sample $(\mathbf{x}, y) \in \mathcal{X} \times \mathcal{Y}$ to a cost. Each client $n$ is assumed to have access to a dataset $\mathcal{D}_n$ which consists of $|\mathcal{D}_n|$ data points sampled from $P_n$. Defining $\bar{\mathcal{L}}_n(\theta) := \frac{1}{|\mathcal{D}_n|} \sum_{(\mathbf{x},y) \in \mathcal{D}_n} \mathcal{L}_n(\theta; (\mathbf{x}, y))$ and $\bar{\mathcal{L}}(\theta) := \frac{1}{N} \sum_{n=1}^{N} \bar{\mathcal{L}}_n(\theta)$, the

optimization problem in FL is $\min_\theta \bar{\mathcal{L}}(\theta)$. The Federated Averaging (FedAvg) (McMahan et al., 2017) algorithm solves this problem by having clients run local stochastic gradient descent (SGD) and send updates to the server, which averages them to update the global model. This cycle repeats until convergence or for a specified number of communication rounds. As the baseline for our proposed approach (detailed in Sec.3), we consider FedAvg, presented as Alg. 3 in App.A.1. This algorithm, unconstrained by privacy limitations, represents the ideal utility case.

**Differential Privacy.** In ML, the mathematical framework of $(\epsilon, \delta)$-DP (Dwork et al., 2014), ensures that two models trained on neighboring datasets, i.e., datasets that differ in one data point, differ only slightly in their outputs. This can be formalized as Def. 1 in App. A.6.2. In $(\epsilon, \delta)$-DP, a smaller privacy budget $\epsilon \in \mathbb{R}_+$ enforces a stronger privacy guarantee. The $\delta \in [0, 1]$ quantifies the probability of violating the privacy guarantee and thereby has usually a small value. We discuss $(\alpha, \epsilon)$-Rényi-DP (Mironov, 2017) as an alternative to relax DP guarantees and get smoother composition in App. A.6.2. In this paper, we adopt RDP, while as it is shown by (Mironov, 2017), it can be converted to DP when needed (cf. Lem. 3 in App. A.6.2).

**DP-FL.** We use the notion of *client-level* DP-FL which protects the entire client's dataset (McMahan et al., 2017; Geyer et al., 2017; Hu et al., 2023). To implement client-level DP in FL, we can rely on the DP-FedAvg (McMahan et al., 2017; Hu et al., 2023) algorithm that first clips clients' updates according to a *sensitivity* $c$ and then adds Gaussian noise according to $\mathcal{N}\left(0, c^2\sigma^2\mathbb{I}\right)$. DP-FedAvg also implements privacy amplification by sub-sampling (Beimel et al., 2014), where clients are sampled independently at random with probability $q$. As shown by Mironov et al. (2019), the Sampled Gaussian Mechanism (SGM) tightens the RDP $\epsilon$ by a quadratic scaling factor $q^2$ (cf. Lem. 4 in App. A.6.2). We also use the notion of *distributed DP* (DDP) (Truex et al., 2019) which combines the advantages of centralized (Ramaswamy et al., 2020) and local DP (Truex et al., 2020). In DDP, clients clip their updates and locally add a small amount of noise, distributed according to $\mathcal{N}\left(0, c^2\sigma^2/N\mathbb{I}\right)$ (Truex et al., 2019). By adding local noise, clients' privacy is partially protected against the server, and sufficient noise, $\mathcal{N}\left(0, c^2\sigma^2\mathbb{I}\right)$, is guaranteed when the server aggregates all $N$ noisy updates. We consider the DP-FedAvg algorithm that is implemented with client-level and DDP (presented as Alg. 4 in App.A.1) as a baseline for our approach, detailed in Sec.3.

**Personalized DP.** DP-FedAvg and most of its variants apply worst-case privacy guarantees to ensure privacy for the most constrained clients. This leads to over-perturbation and reduced utility for clients with more relaxed privacy constraints. Some recent literature addresses this by exploring personalized, or individualized, DP (Yang et al., 2021; Shen et al., 2023; Malekmohammadi et al., 2024; Boenisch et al., 2024). For example, Boenisch et al. (2024) introduce individualized DP (IDP) and apply it to the DP-SGD algorithm, a collaboration-free variant of DP-FedAvg. The integration of IDP into a client-level DP-FL framework is natural. IDP enables individualized privacy budgets across clients and fine-tunes client-specific privacy parameters, through the use of different clip norms and/or sampling rates. We name this integrated algorithm IDP-FedAvg and formally present it as Alg. 5 in App.A.1. We note that IDP-FedAvg and our proposed approach, detailed in Sec.3, complement each other. In this paper, we consider IDP-FedAvg as another baseline for our approach.

**Adaptive DP.** Another approach to improve the utility performance of DP learning is parameter grouping (Yang et al., 2023; McMahan et al., 2017) which clusters the ML parameters with similar clipping norms and applies a non-uniform clipping across clusters. Similar to the IDP approach, our approach and parameter grouping are complementary. However, integrating our approach into parameter grouping optimally, given the increased parameter choices across clusters, requires a study on cluster-based parameter selection which is beyond the scope of this paper. Another approach used for utility improvement is adaptive clipping (Pichapati et al., 2019; Andrew et al., 2021; Li et al., 2022) which aims to optimize the clipping norms during training and thereby reduce the noise effect. Some of these papers either rely on a strong assumption of accessing public data (Li et al., 2022), or the strong assumption of minimal privacy loss occurs during parameter optimization (Pichapati et al., 2019). Our proposed approach is an alternative that selects privacy parameters non-uniformly over time. Compared to the above adaptive clipping methods, our approach eliminates the need for public data, and as parameter selection is done prior to training, ensures zero privacy loss. Another adaptive clipping method that is not limited to prior assumptions and is more relevant to ours is the one proposed by Andrew et al. (2021), and presented as Alg. 6 and detailed in App.A.1. While

targeting the same goal of improving the privacy-utility tradeoff as ours, the method (Andrew et al., 2021) maintains fixed privacy spending over time. Note that adaptive clipping is orthogonal to our approach and could be combined with ours for potential performance gain. However, such integration requires careful privacy analysis, which is beyond the scope of this paper and left for future work. In App. A.8, we benchmark our approach against the method (Andrew et al., 2021), showing that our approach achieves a better privacy-utility tradeoff.

## 3 PROPOSED FRAMEWORK

We introduce a novel time-adaptive DP-FL framework to solve the FL optimization problem with high utility and under privacy constraints that are not meant to be spent uniformly over time. We first discuss our threat space and privacy-related hyperparameters (summarized in Table 1).

**Threat Space.** We assume that each client aims to prevent data leakage to any other client who may be honest but curious (HbC). Specifically, HbC clients follow the FL protocol honesty but may attempt to infer sensitive information of the victim client from the shared model updates. We assume the server is trusted but rather than offering zero protection, we make clients perturb their model updates before sharing with the server to preserve privacy, though at a lower level. This is because, after the server aggregates the perturbed model updates, the total perturbation increases, providing stronger privacy protection against other clients than the server. For enhanced protection against the server, one can use secure aggregation (Bonawitz et al., 2016) which ensures that the server only learns an aggregated function (typically the sum) of the clients' local updates, without learning individual updates. However, the design or implementation of secure aggregation schemes is beyond the scope of this paper, and we mainly focus on preserving privacy against HbC clients.

**Privacy Hyperparameters.** In spend-as-you-go method, which we will detail as part of our framework design in Sec. 3.1, each client saves a specific fraction of their budget in certain rounds, then incrementally spends the saved portion over time. We realize savings by using client-specific sampling rates, denoted as $q_n \in [0, 1]$ for client $n$, during "saving" rounds and by using a uniform, higher, sampling rate, denoted as $q \in [0, 1]$, during "spending" rounds. If $q_n < q$, client $n$ is less likely to be sampled during the saving rounds. According to privacy amplification by sub-sampling (Beimel et al., 2014), and as detailed in Sec. 4.1, a fraction of the clients' privacy budget will thereby be saved for later rounds. Another hyperparameter is each client's designated round for transitioning from the saving to the non-saving (spend) mode. For each $n \in [N]$, we use $T_n$ to denote the first round in which client $n$ is in the spending mode. Intuitively, if $T_n$ is aligned with the client $n$ transitioning from the coarse-grained training of early rounds to the fine-grained feature training of later rounds, the client's saved budget during early rounds enables the client to spend more in later rounds when additional accuracy is beneficial in learning more fine-grained features. By setting $q_n = q$ or $T_n = 1$ for every $n \in [N]$, each client's privacy spending becomes uniform over time, resembling the traditional non-time-adaptive spending approaches. We show the hyper-parameters $q_n$ and $T_n$, which are specific to our framework, in the first two rows of Table 1. Other hyperparameters, which are common in much of the DP-FL literature, are summarized in rows 3 to 10 of Table 1. The privacy-related parameters, consistent across clients and fixed over time, include the global clip norm $c$, the sampling rate $q$, the DP parameter $\delta$, and the RDP-related order $\alpha$. The configuration of these hyperparameters to avoid additional privacy loss is covered partly in simulations (Sec. 5 and App. A.8) and partly in theory (Sec. 4). Later, in Sec. 6, we discuss potential future directions for broader hyperparameter tuning.

### 3.1 THE SPEND-AS-YOU-GO METHOD

We consider an FL setting where every client $n \in [N]$ is given a privacy budget $\epsilon_n$ derived from their individual privacy preferences. We assume all clients exhaust their privacy budget after $T$ global rounds. We propose the spend-as-you-go method, which is executed before training begins, and obtains the following privacy parameters used during execution: the local noise multipliers $\sigma_n^t$, the sampling rates $q_n^t$, the privacy budget remaining $\epsilon_n^t$ (with an initial value of $\epsilon_n^0 = \epsilon_n$), and the local clip norms $c_n^t$. Each $q_n^t$ is chosen as either $q$ or $q_n$, depending on whether client $n$ is in spending or saving mode. From now on, we denote $q_n^t$ as the sampling rate, and to distinguish

between $q$ and $q_n$, we respectively refer to them as the *spending-based* sampling rate and the *saving-based* sampling rate. To denote whether clients are in saving or spending mode at each round, we use $M_n^t$, a binary variable that is set to 0 if round $t$ is a saving round for client $n$, and 1 otherwise. I.e., $M_n^t = 0$ if $t < T_n$ and $M_n^t = 1$ if $t \geq T_n$. Algorithm 1 presents the pseudocode for the spend-as-you-go method. Regardless of the saving or sampling mode, in each round, we first find the value $\sigma_n^t$ required to obtain the remaining privacy budget $\epsilon_n^t$ using the GetNoise[1] function (Line 4 of Alg. 1). This function takes as inputs $\epsilon_n^t$, $\delta$, $q$, and the number of remaining rounds $T - t$. During the spending rounds of client $n$ (i.e., when $M_n^t = 1$), the client is sampled according to $q_n^t = q$. During the saving rounds (when $M_n^t = 0$), the client sets $q_n^t = q_n$. As shown in lines 6-8, we calculate the privacy spent and $\epsilon_n^t$ at the end of each round $t \in [T]$ and for every client $n \in [N]$. Using Compute_rdp function, we calculate the privacy spent given $q_n^t, \sigma_n^t$, and $\alpha$. We then use get_privacy_spent to convert the RDP privacy spent into the DP privacy spent, given the RDP privacy spent, $\alpha$, and $\delta$. We finally follow the same procedure as in Boenisch et al. (2024) to compute local clip norms $c_n^t$ such that their average across $n \in [N]$ equals the hyperparameter $c$.

To achieve this, we calculate $c_n^t = c\sigma^t/\sigma_n^t$ (Line 12), where $\sigma^t = N \left( \sum_{n \in [N]} 1/\sigma_n^t \right)^{-1}$ (Line 10).

Table 1: Hyperparameters Summary

| $T_n$ | Saving-to-spending transition round |
|---|---|
| $q_n$ | Saving-based sampling rate of Client $n$ |
| $T$ | Number of rounds |
| $L$ | Number of local iterations |
| $B$ | Batch size |
| $\lambda$ | Learning rate |
| $\alpha$ | Rényi order in RDP |
| $\delta$ | Probability of violating in DP |
| $c$ | Average clipping norm |
| $q$ | Global (spending-based) sampling rate |

**Algorithm 1** The spend-as-you-go Method in Our Time-adaptive DP-FL Framework

**Inputs:** No. clients $N$, No. global rounds $T$, local privacy budgets $\epsilon_n$, sampling rate $q$, average clip norm $c$, modes $M_n^t \in \{0, 1\}$, sampling rates for saving mode $q_n$, Prob. of violating $\delta$.

**Def** SetPrivacyParams$\left( c, q, \{q_n, \epsilon_n, M_n^t\}_{\substack{n \in [N] \\ t \in [T]}}, T, \delta \right)$

1: **Initialize** $\epsilon_n^0 = \epsilon_n$ and $\bar{\epsilon}_{\text{rdp},n}^0 = 0$ for all $n \in [N]$
2: **for** each global round $t \in [T]$ **do**
3:    **for** each client $n \in [N]$ **do**
4:       $\sigma_n^t$ =GetNoise$\left( \epsilon_n^{t-1}, \delta, q, T-t \right)$
5:       **If** $M_n^t = 0$, **then** $q_n^t = q_n$, **Else**, $q_n^t = q$.
6:       $\epsilon_{\text{rdp},n}^t$ =Compute_rdp$\left( q_n^t, \sigma_n^t, \alpha \right)$
7:       $\bar{\epsilon}_{\text{rdp},n}^t = \bar{\epsilon}_{\text{rdp},n}^{t-1} + \epsilon_{\text{rdp},n}^t$
8:       $\epsilon_n^t = \epsilon_n -$get_privacy_spent$\left( \alpha, \bar{\epsilon}_{\text{rdp},n}^t, \delta \right)$
9:    **end for**
10:   Compute $\sigma^t \leftarrow \left( \frac{1}{N} \sum_{n \in [N]} \frac{1}{\sigma_n^t} \right)^{-1}$
11:   **for** each client $n \in [N]$ **do**
12:    Set local clip norm $c_n^t = \frac{c\sigma^t}{\sigma_n^t}$
13:   **end for**
14:   $q^t = \frac{1}{N} \sum_{n=1}^N q_n^t$
15: **end for**
16: **Return** $\{q^t, \sigma^t, \{q_n^t, c_n^t, \sigma_n^t\}_{n \in [N]}\}_{t \in [T]}$

**Algorithm 2** Iterative Training in Our Time-adaptive DP-FL Framework

**Inputs:** No. clients $N$, No. global rounds $T$, No. local iterations $L$, local privacy budgets $\epsilon_n$, sampling rate $q$, average clip norm $c$, modes $M_n^t \in \{0, 1\}$, sampling rates for saving mode $q_n$, loss functions $\mathcal{L}_n$, datasets $\mathcal{D}_n$, learning rate $\lambda$, batch size $B$, Prob. of violating $\delta$

1: $\{q^t, \sigma^t, \{q_n^t, c_n^t, \sigma_n^t\}_{n \in [N]}\}_{t \in [T]}$
   =SetPrivacyParams$\left( c, q, \{q_n, \epsilon_n, M_n^t\}_{\substack{n \in [N] \\ t \in [T]}}, T, \delta \right)$
2: **Initialize** global model $\theta^0$
3: **for** each global round $t \in [T]$ **do**
4:   $\mathcal{C}^t \leftarrow$ Sample clients with probability $\{q_n^t\}$.
5:   **for** each client $n \in [N]$ in parallel **do**
6:     $\tilde{\Delta}\theta_n^t$ =ClientUpdate$\left( t, n, \theta^{t-1}, c_n^t, \sigma_n^t \right)$.
7:   **end for**
8:   $\tilde{\Delta}\theta^t = \sum_{n \in \mathcal{C}^t} \tilde{\Delta}\theta_n^t + \sum_{n \in [N] \setminus \mathcal{C}^t} \mathcal{N}\left( 0, \frac{c^2(\sigma^t)^2}{N}\mathbb{I} \right)$
9:   Update $\theta^t = \theta^{t-1} + \frac{\tilde{\Delta}\theta^t}{q^t N}$
10: **end for**

**Def** ClientUpdate$\left( t, n, \theta^{t-1}, c_n^t, \sigma_n^t \right)$

1: **Initialize** local model $\theta_n^{t,0} = \theta^{t-1}$
2: **for** local iteration $l \in [l]$ **do**
3:   $\{\mathcal{B}_i\}_{i=1}^{|\mathcal{D}_n|/B} \leftarrow$ Split $\mathcal{D}_n$ to size $B$ batches
4:   **for** each batch $\mathcal{B}_i$ **do**
5:     $\theta_n^{t,l} = \theta_n^{t,l-1} - \frac{\lambda \sum_{(\mathbf{x},y) \in \mathcal{B}_i} \nabla \mathcal{L}_n\left( \theta_n^{t,l-1}; (\mathbf{x},y) \right)}{B}$
6:   **end for**
7: **end for**
8: Compute $\Delta\theta_n^t = \theta_n^{t,L} - \theta_n^{t,0}$
9: Clip $\tilde{\Delta}\theta_n^t = \Delta\theta_n^t \min\left( 1, \frac{c_n^t}{\|\Delta\theta_n^t\|_2} \right)$
10: Add noise $\tilde{\Delta}\theta_n^t \leftarrow \tilde{\Delta}\theta_n^t + \mathcal{N}\left( 0, \frac{(c_n^t)^2(\sigma_n^t)^2}{N}\mathbb{I} \right)$
11: Return $\tilde{\Delta}\theta_n^t$

## 3.2 THE ITERATIVE TRAINING MODULE

After setting parameters through the spend-as-you-go method, our DP-FL framework operates the iterative training module, with the pseudocode presented in Algorithm 2. We note that although the primary contribution of our DP-FL framework lies in the spend-as-you-go method, the iterative

---

[1]To introduce our spend-as-you-go method, we use some functions as implemented by Opacus library (Yousefpour et al., 2021).

training module also differs from the baseline due to the use of time-adaptive privacy parameters and has to be carefully designed. This module spans $T$ communication rounds. Within each round, clients conduct $L$ iterations of local training. For iteration $l \in [L]$ within round $t \in [T]$, let $\theta_n^{t,l}$ denote the local model of client $n$. In round $t \in [T]$, $\mathcal{C}^t$ denotes the set of workers contributing to local training. This set is randomly chosen using Poisson sampling (Line 4). Each client $n \in [N]$ in round $t \in [T]$ has an independent probability $q_n^t \in [0,1]$ of being selected for $\mathcal{C}^t$. In expectation $q^t N$ clients, where $q^t = \frac{1}{N} \sum_{n \in [N]} q_n^t$, are sampled in round $t$.

**Client update**: In round $t$, each client $n \in [N]$ initializes $\theta_n^{t,0} = \theta^{t-1}$ (Line 1 of `ClientUpdate`). It then performs $L$ iterations of local training, using a gradient-based technique, such as mini-batch SGD. In each iteration $l \in [L]$, the client first splits $\mathcal{D}_n$ into size $B$ batches (Line 3 of `ClientUpdate`). For each batch $\mathcal{B}_i$, the $n$th client updates $\theta_n^{t,l} = \theta_n^{t,l-1} - \frac{\lambda}{B} \sum_{(\mathbf{x},y) \in \mathcal{B}_i} \nabla \mathcal{L}_n \left( \theta_n^{t,l-1}; (\mathbf{x},y) \right)$, where $\lambda$ is the learning rate and $\frac{1}{B} \sum_{(\mathbf{x},y) \in \mathcal{B}_i} \nabla \mathcal{L}_n \left( \theta_n^{t,l-1}; (\mathbf{x},y) \right)$ estimates the gradient $\nabla \bar{\mathcal{L}}_n(\theta_n^{t,l-1})$ (Line 5 of `ClientUpdate`). Once local training finishes, the client computes the resulting model update $\Delta \theta_n^t = \theta_n^{t,L} - \theta_n^{t,0}$ (Line 8 of `ClientUpdate`). The first phase of integrating the DP mechanism in the iterative training module is to assign the client the task of clipping $\Delta \theta_n^t$ as shown in (Line 9 of `ClientUpdate`), and detailed as $\tilde{\Delta} \theta_n^t = \Delta \theta_n^t \min \left( 1, \frac{c_n^t}{\|\Delta \theta_n^t\|_2} \right)$, where $c_n^t$ is the clip norm. The client then perturbs further its clipped model update by injecting random noise (Line 10 of `ClientUpdate`). The noise is selected independently for each client $n \in \mathcal{C}^t$ in round $t$. Algebraically, given noise $z_n^t \sim \mathcal{N} \left( 0, \frac{(c_n^t \sigma_n^t)^2}{N} \mathbb{I} \right)$, $\tilde{\Delta} \theta_n^t \leftarrow \tilde{\Delta} \theta_n^t + z_n^t$.

**Server aggregation:** As shown in lines 6 and 8, the server aggregates $\tilde{\Delta} \theta_n^t$ for all $n \in \mathcal{C}^t$, and computes the sum $\tilde{\Delta} \theta^t = \sum_{n \in \mathcal{C}^t} \tilde{\Delta} \theta_n^t$. For those clients not being selected, i.e., $n \in [N] \backslash \mathcal{C}^t$, the server compensates by injecting additional noise $\tilde{\Delta} \theta^t \leftarrow \tilde{\Delta} \theta^t + \sum_{n \in [N] \backslash \mathcal{C}^t} \mathcal{N} \left( 0, c^2 (\sigma^t)^2 / N \mathbb{I} \right)$. Assuming $c \sigma^t = c_n^t \sigma_n^t$ for all $n \in [N]$, because of this compensation, the total noise power $\mathcal{N} \left( 0, c^2 (\sigma^t)^2 \mathbb{I} \right)$ that is injected to the global model update is not a function of the sampling. The round ends with the server computing $\theta^t = \theta^{t-1} + \frac{\tilde{\Delta} \theta^t}{q^t N}$ (Line 9).

## 4 THEORETICAL ANALYSIS

For the DP-FL framework (described in Sec. 3), we now develop the theory that underlies our privacy accounting (Sec. 4.1), and optimize the sampling rates to improve utility (Sec. 4.2).

### 4.1 PRIVACY ACCOUNTING

In this section, we calculate the RDP bounds for each client $n \in [N]$ at round $t \in [T]$. The results will support the general idea of save-to-spend in our framework. Since we use RDP to perform privacy accounting, we convert each client's privacy budgets $\{\epsilon_n\}_{n \in [N]}$ into the RDP equivalent, denoted $\{\epsilon_{\text{rdp},n}\}_{n \in [N]}$, at a fixed order $\alpha$. For client $n$, we use $\epsilon_{\text{rdp},n}^t$ and $\epsilon_{\text{rdp-left},n}^t$ to denote, respectively, the RDP privacy spent in round $t$ and the "go-forward" RDP privacy budget remaining for round $t$ onwards.

The privacy accounting for the spend-as-you-go method first involves calculating each $\epsilon_{\text{rdp},n}^t$ (cf. line 6 of Alg. 1). Next, the total RDP privacy spent during the first $t$ rounds is calculated as $\sum_{\tau=1}^{t-1} \epsilon_{\text{rdp},n}^\tau$ (Line 7 of Alg. 1), with budget remaining $\epsilon_{\text{rdp-left},n}^t = \epsilon_{\text{rdp},n} - \sum_{\tau=1}^{t-1} \epsilon_{\text{rdp},n}^\tau$. Recalling from Sec. 3.1, client $n$ selects the noise multiplier $\sigma_n^t$ assuming that $\epsilon_{\text{rdp-left},n}^t$ will be spent uniformly across the remaining $T - t + 1$ rounds subject to using sampling rate $q$. As shown by Mironov et al. (2019), under this uniform assumption $\sigma_n^t$ should satisfy $\frac{\epsilon_{\text{rdp-left},n}^t}{T-t+1} = \frac{2\alpha q^2}{\sigma_n^t}$. When $t < T_n$, the sampling rate $q_n$ reduces the RDP expenditure to $\epsilon_{\text{rdp},n}^t = \frac{\epsilon_{\text{rdp-left},n}^t (q_n)^2}{(T-t+1)(q)^2}$. When $t \geq T_n$, the full allocated budget is used in round $t$. The RDP privacy spend $\epsilon_{\text{rdp},n}^t$ can be computed recursively as

$$\epsilon_{\text{rdp},n}^t = \left( \frac{\epsilon_{\text{rdp},n} - \sum_{\tau=1}^{t-1} \epsilon_{\text{rdp},n}^\tau}{T-t+1} \right) \left( \mathbf{1}_{t < T_n} \left( \frac{q_n^t}{q} \right)^2 + \mathbf{1}_{t \geq T_n} \right). \tag{1}$$

Lemma 1 solves the recursive formula (1) for the RDP spent. Theorem 1 shows the RDP spent is non-decreasing over time. The proofs of Lem. 1 and Thm. 1 are respectively provided in App. A.2 and A.3.

**Lemma 1.** *Given any $n \in [N], t \in [T]$, and $T_n \in [T]$, we have*

$$
\epsilon_{rdp,n}^t = \begin{cases} \frac{\epsilon_{rdp,n}}{T-t+1} \left(\frac{q_n}{q}\right)^2 \prod_{i=1}^{t-1} \left(1 - \frac{1}{T-t+1+i} \left(\frac{q_n}{q}\right)^2\right) & \text{if } t < T_n \\ \frac{\epsilon_{rdp,n}}{T-T_n+1} \prod_{i=1}^{T_n-1} \left(1 - \frac{1}{T-T_n+1+i} \left(\frac{q_n}{q}\right)^2\right) & \text{ow} \end{cases}.
\tag{2}
$$

**Theorem 1.** *For $(n,t,T_n) \in [N] \times [T] \times [T]$, $\epsilon_{rdp,n}^t \geq \epsilon_{rdp,n}^{t-1}$ if $t \leq T_n$, and $\epsilon_{rdp,n}^t = \epsilon_{rdp,n}^{t-1}$ if $t > T_n$.*

**Remark 1.** *The non-decreasing result of Thm. 1 indicates that during saving rounds ($M_n^t = 0$) clients spend at least as much of their privacy budget in each round as in the previous round. In other words, saving decreases over time. During spending rounds ($M_n^t = 1$) clients expend privacy budget at a constant rate. If budget savings are accumulated than $\epsilon_{rdp,n}^{T_n} > \epsilon_{rdp,n}^{T_n-1}$ and clients will have access to a larger budget to spend.*

## 4.2 Optimal Permutation of Saving-based Sampling Rates

We now optimize the selection of sampling rates. We start from a pre-defined set of $N$ sampling rates. We then choose a permutation that assigns each of the $N$ rates to a distinct client. The permutation is selected to minimize the per-round difference between the utility achieved when DP perturbation is not applied (which generally will maximize utility), and the utility achieved when our DP-FL framework is used. In the first case, the server aggregates the unperturbed local updates $\Delta\theta_n^t$ (i.e., no clipping or noise addition) for all clients $n \in [N]$ (i.e., no sub-sampled), and updates the global model as $\theta^t = \theta^{t-1} + \Delta\theta^t$ where

$$
\Delta\theta^t = \frac{1}{N} \sum_{n=1}^{N} \Delta\theta_n^t.
\tag{3}
$$

In our DP-FL framework, local updates undergo the DP mechanism detailed in Sec. 3. Factoring into the (modified) global update $\tilde{\Delta}\theta^t$ the clipping norms $c_n^t$, the additive noise multipliers $\sigma_n^t$, and the sampling rates $q_n^t$, we get

$$
\tilde{\Delta}\theta^t = \frac{1}{Nq^t} \sum_{n=1}^{N} \left( b_n^t \left( \text{Clip}\left(\Delta\theta_n^t, c_n^t\right) + z_n^t \right) + (1 - b_n^t)\tilde{z}_n^t \right),
\tag{4}
$$

where $q^t = \frac{1}{N} \sum_{n=1}^{N} q_n^t$, $z_n^t, \tilde{z}_n^t \sim \mathcal{N}\left(0, \frac{(\sigma_n^t c_n^t)^2}{N}\right)$ are identically and independently distributed (IID), and $b_n^t \sim \text{Bern}(q_n^t)$ are also IID. The optimization problem is to choose the $\{q_n^t\}_{n \in [N]}$ so that $\tilde{\Delta}\theta^t$ closely approximates an unbiased estimate of $\Delta\theta^t$. We define the difference between the respective model updates as

$$
\text{Error}^t := \Delta\theta^t - \tilde{\Delta}\theta^t.
\tag{5}
$$

$\text{Error}^t$ has four sources of randomness:

- (i) **Local dataset randomness**: the randomness of local datasets $\mathcal{D}_n$, which are sampled from distributions $P_n$. This randomness is reflected in the local updates $\Delta\theta_n^t$.
- (ii) **Client sampling randomness:** the sampling of clients, represented by the use of random variables $b_n^t$. These determine whether a client $n$ contributes to the round $t$'s local training.
- (iii) **Noise addition randomness:** the addition of the Gaussian noises $z_n^t$ and $\tilde{z}_n^t$.
- (iv) **Privacy budget assignment randomness:** the matching of clients with different datasets $\mathcal{D}_n$ and data distributions $P_n$ to different privacy budgets $\epsilon_n$. Here, $\{\epsilon_n\}_{n \in [N]}$ are considered as a random permutation of a predefined set of privacy budgets $\{\hat{\epsilon}_n\}_{n \in [N]}$.

To optimize the sampling rates, we first develop two upper bounds on the bias term $\left\|\mathbb{E}\left(\text{Error}^t\right)\right\|$. Both bounds build on the clipping bias lemma (Das et al., 2023), cf., Lem. 2 in App. A.6.1. Our theorems extend the results of that lemma to the situation where clients have individualized privacy budgets and use time-varying clip norms $c_n^t$ and sampling rates $q_n^t$.

Our first theorem, Thm. 2, bounds the expected bias with respect to (w.r.t.) three sources of randomness: (i), (ii), and (iii). We denote this expectation as $\left\|\mathbb{E}_{(i),(ii),(iii)}\left(\text{Error}^t\right)\right\|$. The proof of Thm. 2 is given in App. A.4.

**Theorem 2.** *Taking the expectation w.r.t. (i), (ii), and (iii), and for any $\rho > 1$, we have*

$$\left\|\mathbb{E}_{(i),(ii),(iii)}\left(\textit{Error}^t\right)\right\| \leq \frac{1}{N}\left\|\sum_{n=1}^{N}\left(1 - \frac{q_n^t}{q^t}\right)\mathbb{E}_{(i)}\left(\Delta\theta_n^t\right)\right\| + \frac{1}{N}\sum_{n=1}^{N}\frac{q_n^t}{q^t}\frac{\mathbb{E}_{(i)}\left(\|\Delta\theta_n^t\|^\rho\right)}{\left(c_n^t\right)^{\rho-1}}. \quad (6)$$

To minimize this bias term in a reasoned fashion we select the $q_n^t$ to minimize the upper bound. The upper bound in (6) contains terms that couple $q_n^t$ with the pure local updates $\Delta\theta_n^t$. The latter are not accessible to the server who is responsible for sampling clients. If clients were to select their own $q_n^t$ based on their local updates, this could lead to privacy leakage and would require additional privacy protection. The coupling between $q_n^t$ and $\Delta\theta_n^t$ can be removed from the first term of the upper bound in (6) under certain conditions. For example, if all clients are sampled at the same rate ($q_n^t = q^t$) or if $\mathbb{E}_{(i)}\left(\Delta\theta_n^t\right)$ is equal across all $n \in [N]$, the first term becomes zero. In such cases, the upper bound reduces to the second term, which still couples $q_n^t$ with $\Delta\theta_n^t$ and the clip norms $c_n^t$.

In contrast to Thm. 2, in Thm. 3 we take the expectation w.r.t. the additional source of randomness, (iv). This results in a bound that depends solely on clipping norms, which makes optimizing the sampling rates, $q_n^t$, easier to accomplish. As we now bound the expected bias w.r.t. all four sources of randomness – (i), (ii), (iii), and (iv) – we denote the expectation as $\left\|\mathbb{E}_{(i),(ii),(iii),(iv)}\left(\text{Error}^t\right)\right\|$. The proof of Thm. 3 is given in App. A.5.

**Theorem 3.** *Taking the expectation w.r.t. (i), (ii), (iii), and (iv), and for any $\rho > 1$, we have:*

$$\left\|\mathbb{E}_{(i),(ii),(iii),(iv)}\left(\textit{Error}^t\right)\right\| \leq \frac{1}{N^2}\left(\sum_{n=1}^{N}\mathbb{E}_{(i)}\left(\|\Delta\theta_n^t\|^\rho\right)\right)\sum_{n=1}^{N}\left(\frac{q_n^t}{q^t}\frac{1}{\left(c_n^t\right)^{\rho-1}}\right). \quad (7)$$

The main step in the proofs of Thm. 3 builds on the common assumption in FL that the sampling from $P_n$ and of $\epsilon_n$ is independent. This assumption is made without significant loss of generality as privacy budgets are often assigned based on clients' personal preferences and policy requirements. In contrast, the data distribution is influenced by external factors such as geographical locations. Such decoupling of privacy budgets and data distributions simplifies the proof of Thm. 3. It avoids the need to model potential correlations between the client's data and privacy preferences. These are often unknown or irrelevant in practice.

As per Thm. 3, to minimize $\left\|\mathbb{E}_{(i),(ii),(iii),(iv)}\left(\text{Error}^t\right)\right\|$ w.r.t. the sampling rates $q_n^t$, we solve the following optimization problem:

$$\min_{\{\Pi_t\}_{t=1}^T}\quad\sum_{n=1}^{N}\frac{q_n^t}{q^t}\frac{1}{\left(c_n^t\right)^{\rho-1}}\quad, \quad (8)$$
$$\text{s.t.}\quad q_n^t = q_{\Pi_t^{-1}(n)}, n \in [N]$$

where the $\{\Pi_t\}$ are a set of (bijective) permutation maps $\Pi_t : [N] \to [N]$. Each permutation maps each element of the index set $[N]$ to a distinct element of $[N]$. We apply the permutation to the indices of the fixed set of sampling rates $\{q_1, \ldots, q_N\}$ to get $\{q_1^t, \ldots, q_N^t\}$. The set $\{q_1, \ldots, q_N\}$ is a hyperparameter in our problem, which is constrained by the condition $q_n \leq q$ for every $n \in [N]$. The optimized choice of the set of sampling rates $\{q_1, \ldots, q_N\}$ is reserved for future work. Per (8), the optimal choice for $\{q_1^t, \ldots, q_N^t\}$ is to assign clients with smaller clip norms $c_n^t$ (which will contribute highly perturbed model updates) to have lower sampling rates $q_n^t$, and vice versa. The intuition is that by matching the $q_n^t$ to the $c_n^t$, we prevent clients who contribute a highly perturbed model update from deteriorating other clients' performance during saving rounds. This reduces clipping bias and allows these clients to preserve more of their privacy budget compared, with the subsequent benefit of enabling them to contribute more in future rounds.

## 5 EXPERIMENTS

**Datasets.** To empirically evaluate the performance of our framework against baselines, we consider widely used datasets. For the Fashion MNIST (FMNIST) and MNIST, we use a convolutional neural network (CNN) architecture from (McMahan et al., 2017). For the Adult Income dataset, we use multi-layer perception from (Zhang et al., 2020). For the CIFAR10 dataset, we use a CNN architecture from He et al. (2016). We partition datasets across 100 clients in a non-IID manner using the Dirichlet distribution with a default parameter 0.1 (Zhang et al., 2023).

**Privacy Settings.** To reduce the number of choices for clients' privacy budgets, and motivated by society wherein individuals often share similar privacy preferences (Alaggan et al., 2015; Boenisch et al., 2024), in our experiments we divide clients into three groups. The groups are respectively assigned budgets of $\epsilon_{\text{group},1}$, $\epsilon_{\text{group},2}$, or $\epsilon_{\text{group},3}$. We randomly allocate $34\%$ of clients to belong to Group 1, $43\%$ to Group 2, and $23\%$ to Group 3. In other words, Client $n$ in Group $m$, shares $\epsilon_n = \epsilon_{\text{group},m}$. To account for privacy consumption, we use the Opacus library (Yousefpour et al., 2021). We consider DP parameter $\delta = 10^{-5}$ and choose an extended version of the default RDP parameter $\alpha$ from the RDPAccountant function in Opacus. We apply per-layer clipping (McMahan et al., 2017) to restrict the influence of individual layers by constraining their norms.

**Baselines.** As discussed in Sec. 2, we consider several baselines. The first is the non-private FedAvg (McMahan et al., 2017) which represents our upper bound on utility without any privacy constraints. The second is DP-FedAvg (McMahan et al., 2017) where we use a uniform privacy budget, chosen according to the smallest epsilon value of any of the clients. This ensures that no client's privacy budget is exceeded. The third is IDP-FedAvg which is the IDP-integration (Boenisch et al., 2024) of DP-FedAvg. The fourth is adaptive clipping (Andrew et al., 2021). To have a fair comparison with the baselines, we evaluate two variants of our framework. The first variant assumes every client's budget is constrained by the smallest value in the group budget tuple $(\epsilon_{\text{group},1}, \epsilon_{\text{group},2}, \epsilon_{\text{group},3})$. The second variant incorporates different privacy groups. Further details on the choice of hyperparameters in our experiments are given in Table 4 in Appendix A.7.

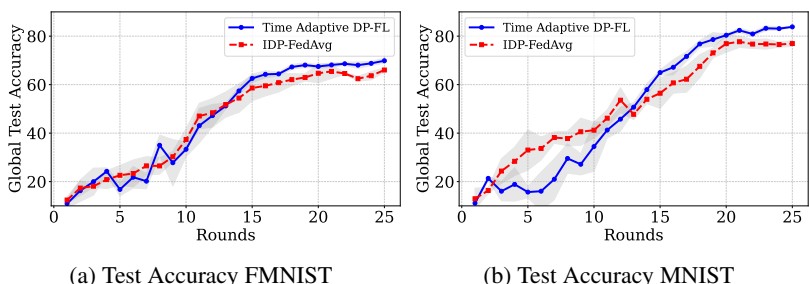

(a) Test Accuracy FMNIST        (b) Test Accuracy MNIST

Figure 1: **Our framework improves accuracy in later rounds compared to the baseline.** We plot the global test accuracy vs. rounds for (a) the FMNIST dataset, and (b) the MNIST dataset. In (a), $(\epsilon_{\text{group},1}, \epsilon_{\text{group},2}, \epsilon_{\text{group},3}) = (10, 20, 30)$, and in (b) it equals $(10, 15, 20)$.

Table 2: **Global test accuracy** for FedAvg without DP constraints, DP-FedAvg with $\epsilon_n = 10$, IDP-FedAvg with non-uniform privacy budgets, our framework with $\epsilon_n = 10$, and our framework with non-uniform budgets. For the FMNIST and Adult Income datasets, the non-uniform privacy budgets $\epsilon_n$ are $(\epsilon_{\text{group},1}, \epsilon_{\text{group},2}, \epsilon_{\text{group},3}) = (10, 20, 30)$, and for MNIST, they are $(10, 15, 20)$.

| DATASET | FedAvg (non-DP) | DP-FedAvg $\epsilon_n = 10$ | IDP-FedAvg Non-uniform | Ours $\epsilon_n = 10$ | Ours Non-uniform |
|---|---|---|---|---|---|
| FMNIST | 72.95 | 64.8 | 65.45 | **67.90** | **70.57** |
| MNIST | 90.23 | 76.79 | 76.94 | **80.2** | **83.83** |
| Adult Income | 78.93 | 60.12 | 70.93 | **72.14** | **77.53** |

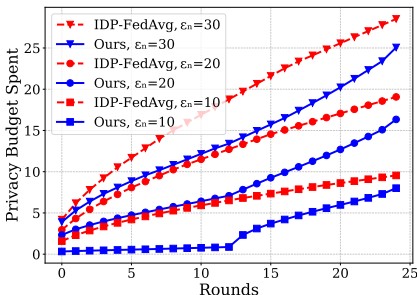

Figure 2: **While both adhere to privacy budgets, our framework follows spend-as-you-go, whereas IDP-FedAvg uses uniform privacy spending.** The blue solid curves correspond to clients' privacy spending in our framework, while the red dashed curves show IDP-FedAvg. The curves of clients with budgets of 30, 20, and 10 are marked with rectangles, circles, and squares, respectively.

**Experimental Results.** As shown in Table 2 and Fig. 1, our framework yields improvements in the resulting global model's accuracy by spending privacy budget non-uniformly across training rounds. Comparing Columns 4 and 6 of Table 2, our framework with non-uniform privacy budgets improves global test accuracy over IDP-FedAve by 7.8%, 8.9%, and 9.3% on FMNIST, MNIST, and Adult Income. In the case of using a uniform budget of $\epsilon_n = 10$ across all clients $n$, our framework achieves respective improvements of 4.7%, 4.4%, and 19.9% compared to DP-FedAvg, as shown in Columns 3 and 5. We also observe that, our time-adaptive DP-FL scheme comes closest to the ideal-case performance of FedAvg (column 2) without privacy constraints. Figure 1 plots global test accuracy vs. global rounds for our framework with non-uniform privacy budgets and IDP-FedAvg, on the FMNIST and MNIST datasets. This figure shows that while our framework conserves privacy in early rounds, it allocates more budget in later rounds, eventually catching up to and surpassing IDP-FedAvg by about 8% on FMNIST and 6% on MNIST in the final round.

In Fig. 2 we present the privacy budget spent by clients from different budget groups $(10, 20, 30)$ across rounds. This figure shows that, while IDP-FedAvg enforces uniform privacy consumption over time, in our framework, clients follow spend-as-you-go, saving budgets in the first half of training, and spending more in later rounds. Our experimental results demonstrate that our time-adaptive approach boosts the utility of the trained model while adhering to privacy constraints.

In Appendix A.8, we present extended experimental results, including benchmarks on the CIFAR10 dataset (Table 11), comparisons with adaptive clipping (Table 7), and evaluations across privacy-related hyperparameters (Tables 6, 9, and 10, and Figure 5), as well as other parameters (Tables 5 and 8 and Figure 4).

## 6 DISCUSSIONS AND FUTURE WORK

We now discuss some limitations of our work that represent interesting directions for future work. Our spend-as-you-go method reduces reliance on determining when clients should transition from saving to spending by allowing them to gradually spend their saved budgets over time, rather than waiting until a specific round to start spending. While our experiments indicate that transitioning from saving to spending midway through training generally yields good results, tuning the hyperparameters involved in estimating the transitioning round may improve utility. However, such hyperparameter tuning can lead to additional privacy loss (Papernot & Steinke, 2021) that would need to be accounted for. For future work, we believe that our time-adaptive DP-FL framework should be closely integrated with a form of privacy-preserving hyperparameter tuning to identify the best rounds in which to transition from savings to spending.

Furthermore, as we demonstrated theoretically and validated experimentally, adapting saving-related hyperparameters to clients' specific privacy budgets can enhance utility. To eliminate the risk of privacy leakage from this adaptation, we provide theoretical optimizations that rely solely on clients' privacy-related constraints, independent of their data. Future research can explore data-and-privacy joint measures to quantify clients' contributions with controlled privacy leakage and adapt client-specific savings decisions accordingly.

ACKNOWLEDGMENTS

We would like to acknowledge our sponsors. This work was supported in part by a Discovery Research Grant from the Natural Sciences and Engineering Research Council of Canada (NSERC), by an NSERC Alexander Graham Bell Canada Graduate Scholarship-Doctoral (CGS D3), by a DiDi graduate award, and by the Mitacs Globalink research award.

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

# A APPENDIX

## A.1 SUMMARY OF NOTATIONS AND BENCHMARKING SCHEMES

We summarize the important notations (including the hyperparameters shown in Table 1) in Table 3. We formally depict the baselines: FedAvg (McMahan et al., 2017) as Alg. 3, DP-FedAvg as Alg. 4, IDP-FedAvg as Alg. 5, and adaptive clipping (Andrew et al., 2021) as Alg. 6. Next, we provide further details to explain the adaptive clipping baseline in comparison with our proposed approach.

Table 3: A Summary of Notation and Hyperparameters[1].

| | | | |
|---|---|---|---|
| $N$ | No. of Clients | $\mathcal{D}_n$ | Dataset of Client $n$ |
| $\mathcal{C}^t$ | Client set in R. $t$ | $\mathcal{B}_i$ | Batch $i$ |
| $T$ | No. of rounds | $B$ | Batch size |
| $L$ | No. of local iterations | $\epsilon_n$ | DP privacy budget of Client $n$ |
| $\theta^t$ | Global model at R. $t$ | $\epsilon_{\text{rdp},n}^t$ | RDP privacy spent of Client $n$ in R. $t$ |
| $\Delta\theta^t$ | Global model update at R. $t$ | $\epsilon_{\text{rdp-left},n}^t$ | RDP budget RE. of Client $n$ for R. $t$ onwards |
| $\theta_n^{t,l}$ | Model of Client $n$ at R. $t$, I. $l$ | $\bar{\epsilon}_{\text{rdp},n}^t$ | RDP privacy spend of Client $n$ up to R. $t+1$ |
| $\Delta\theta_n^t$ | Model update of Client $n$ at R. $t$ | $\epsilon_n^t$ | DP budget RE. of Client $n$ for R. $t$ onwards |
| $\tilde{\Delta}\theta_n^t$ | Perturbed update of Client $n$ at R. $t$ | $\sigma^t$ | Global noise multiplier in R. $t$ |
| $\text{Error}_t$ | $\Delta\theta^t - \tilde{\Delta}\theta^t$ | $\sigma_n^t$ | Noise multiplier of Client $n$ in R. $t$ |
| $\lambda$ | Learning rate | $c$ | Average clipping norm |
| $\alpha$ | Rényi order in RDP | $c_n^t$ | Clipping norm of Client $n$ in R. $t$ |
| $\delta$ | Probability of violating in DP | $q$ | Spending-based sampling rate |
| $T_n$ | Saving-to-spending transition R. | $q_n$ | Saving-based sampling rate of Client $n$ |
| $M_n^t$ | Saving-or-spending mode | $q_n^t$ | Sampling rate of Client $n$ in R. $t$ |
| $\mathcal{L}_n$ | Loss function of Client $n$ | $q^t$ | Average sampling rate in R. $t$ |

[1] Table's abbreviations: "No." for "Number", "RE." for "Remaining", "I." for "Iteration", and "R." for "Round".

---

**Algorithm 3** Federated Averaging (FedAvg) (McMahan et al., 2017)

**Inputs:** No. clients $N$, No. global rounds $T$, No. local iterations $L$, loss functions $\mathcal{L}_n$, local datasets $\mathcal{D}_n$, learning rate $\lambda$, batch size $B$

1: **Initialize** global model $\theta^0$
2: **for** each global round $t \in [T]$ **do**
3:    $\mathcal{C}^t \leftarrow$ Sample clients with probability $q$.
4:    **for** each client $n \in \mathcal{C}^t$ in parallel **do**
5:       $\Delta\theta_n^t = \text{ClientUpdate}(t, n, \theta^{t-1})$.
6:    **end for**
7:    Aggregate $\Delta\theta^t = \sum_{n\in\mathcal{C}^t} \tilde{\Delta}\theta_n^t$
8:    Update $\theta^t = \theta^{t-1} + \frac{\Delta\theta^t}{qN}$
9: **end for**

**Def** $\text{ClientUpdate}(t, n, \theta^t)$

1: **Initialize** local model $\theta_n^{t,0} = \theta^t$
2: **for** local iteration $l \in [l]$ **do**
3:    $\{\mathcal{B}_i\}_{i=1}^{|\mathcal{D}_n|/B} \leftarrow$ Split $\mathcal{D}_n$ to size $B$ batches
4:    **for** each batch $\mathcal{B}_i$ **do**
5:       $\theta_n^{t,l} = \theta_n^{t,l-1} - \frac{\lambda \sum_{(\mathbf{x},y)\in\mathcal{B}_i} \nabla\mathcal{L}_n(\theta_n^{t,l-1};(\mathbf{x},y))}{B}$
6:    **end for**
7: **end for**
8: Return $\Delta\theta_n^t = \theta_n^{t,L} - \theta_n^{t,0}$

---

**Algorithm 4** Differential Private Federated Averaging (DP-FedAvg) (McMahan et al., 2017)

**Inputs:** No. clients $N$, No. global rounds $T$, No. local iterations $L$, noise multiplier $\sigma$, clip norm $c$, sampling rate $q$, loss functions $\mathcal{L}_n$, local datasets $\mathcal{D}_n$, learning rate $\lambda$, batch size $B$

1: **Initialize** global model $\theta^0$
2: **for** each global round $t \in [T]$ **do**
3:    $\mathcal{C}^t \leftarrow$ Sample clients with probability $q$.
4:    **for** each client $n \in \mathcal{C}^t$ in parallel **do**
5:       $\tilde{\Delta}\theta_n^t = \text{ClientUpdate}(t, n, \theta^{t-1}, c)$.
6:    **end for**
7:    Aggregate $\tilde{\Delta}\theta^t = \sum_{n\in\mathcal{C}^t} \tilde{\Delta}\theta_n^t$
8:    Add noise $\tilde{\Delta}\theta^t \leftarrow \tilde{\Delta}\theta^t + \mathcal{N}(0, c^2\sigma^2\mathbb{I})$
9:    Update $\theta^t = \theta^{t-1} + \frac{\tilde{\Delta}\theta^t}{qN}$
10: **end for**

**Def** $\text{ClientUpdate}(t, n, \theta^t, c)$

1: **Initialize** local model $\theta_n^{t,0} = \theta^t$
2: **for** local iteration $l \in [l]$ **do**
3:    $\{\mathcal{B}_i\}_{i=1}^{|\mathcal{D}_n|/B} \leftarrow$ Split $\mathcal{D}_n$ to size $B$ batches
4:    **for** each batch $\mathcal{B}_i$ **do**
5:       $\theta_n^{t,l} = \theta_n^{t,l-1} - \frac{\lambda \sum_{(\mathbf{x},y)\in\mathcal{B}_i} \nabla\mathcal{L}_n(\theta_n^{t,l-1};(\mathbf{x},y))}{B}$
6:    **end for**
7: **end for**
8: Compute $\Delta\theta_n^t = \theta_n^{t,L} - \theta_n^{t,0}$
9: Clip $\tilde{\Delta}\theta_n^t = \Delta\theta_n^t \min\left(1, \frac{c}{\|\Delta\theta_n^t\|_2}\right)$
10: Return $\tilde{\Delta}\theta_n^t$

**The Adapting Clipping Baseline**. In this paper, we consider the adaptive clipping method (Andrew et al., 2021) as a baseline for our time-adaptive DP-FL approach. In the extended simulations (cf. App. A.8), we benchmark that method against our approach. The method is formally presented as Alg. 6. As shown in Line 13, the server dynamically adjusts the clipping norm based on a specified quantile $\gamma$ of the distribution of clients' updates. The goal of this method is to minimize the difference between the clipping norm and the quantile in the distribution, aiming to achieve the same objective as ours: improving the privacy-utility tradeoff. In contrast to our time-adaptive approach, which is independent of the client's data and can be done prior to training, the method (Andrew et al., 2021) introduces privacy risks during the quantile approximation. To mitigate these risks, and as is shown in Line 2 of the `SetClipping` function in Alg. 6, the method (Andrew et al., 2021) incorporates a supplementary DP mechanism that allocates part of the privacy budget to preserve privacy during quantile estimation. However, this results in a lower remaining privacy budget, requiring a larger noise multiplier $\sigma$, as computed in Line 2 of `SetSigma` in Alg.6, in comparison to our approach.

---

**Algorithm 5** Individualized DP-FedAvg (IDP-FedAvg), a natural integration of IDP (Boenisch et al., 2024) to FL

**Inputs:** No. clients $N$, No. global rounds $T$, No. local iterations $L$, local privacy budgets $\epsilon_n$, average clip norm $c$, sampling rate $q$, loss functions $\mathcal{L}_n$, local datasets $\mathcal{D}_n$, learning rate $\lambda$, batch size $B$, probability of violating $\delta$

1: $\sigma, \{c_n\}_{n\in[N]} =\text{SetPrivacyParams}(c, q, \{\epsilon_n\}_{n\in[N]}, T, \delta)$
2: **Initialize** global model $\theta^0$
3: **for** each global round $t \in [T]$ **do**
4: $\quad \mathcal{C}^t \leftarrow$ Sample clients with probability $q$.
5: $\quad$ **for** each client $n \in \mathcal{C}^t$ in parallel **do**
6: $\quad\quad \tilde{\Delta}\theta_n^t = \text{ClientUpdate}(t, n, \theta^{t-1}, c_n)$.
7: $\quad$ **end for**
8: $\quad$ Aggregate $\tilde{\Delta}\theta^t = \sum_{n\in\mathcal{C}^t} \tilde{\Delta}\theta_n^t$
9: $\quad$ Add noise $\tilde{\Delta}\theta^t \leftarrow \tilde{\Delta}\theta^t + \mathcal{N}(0, c^2\sigma^2\mathbb{I})$
10: $\quad$ Update $\theta^t = \theta^{t-1} + \frac{\tilde{\Delta}\theta^t}{qN}$
11: **end for**

**Def** `ClientUpdate`$(t, n, \theta^t, c_n)$

1: **Initialize** local model $\theta_n^{t,0} = \theta^t$
2: **for** local iteration $l \in [l]$ **do**
3: $\quad \{\mathcal{B}_i\}_{i=1}^{|\mathcal{D}_n|/B} \leftarrow$ Split $\mathcal{D}_n$ to size $B$ batches
4: $\quad$ **for** each batch $\mathcal{B}_i$ **do**
5: $\quad\quad \theta_n^{t,l} = \theta_n^{t,l-1} - \frac{\lambda \sum_{(\mathbf{x},y)\in\mathcal{B}_i} \nabla\mathcal{L}_n(\theta_n^{t,l-1};(\mathbf{x},y))}{B}$
6: $\quad$ **end for**
7: **end for**
8: Compute $\Delta\theta_n^t = \theta_n^{t,L} - \theta_n^{t,0}$
9: Clip $\tilde{\Delta}\theta_n^t = \Delta\theta_n^t \min\left(1, \frac{c_n}{\|\Delta\theta_n^t\|_2}\right)$
10: Return $\tilde{\Delta}\theta_n^t$

**Def** `SetPrivacyParams`$(c, q, \{\epsilon_n\}_{n\in[N]}, T, \delta)$

1: **for** each client $n \in [N]$ **do**
2: $\quad$ Set local noise multiplier $\sigma_n =\text{GetNoise}(\epsilon_n, \delta, q, T)$
3: **end for**
4: Compute $\sigma \leftarrow \left(\frac{1}{N}\sum_{n\in[N]} \frac{1}{\sigma_n}\right)^{-1}$
5: **for** each client $n \in [N]$ **do**
6: $\quad$ Set local clip norm $c_n = \frac{c\sigma}{\sigma_n}$
7: **end for**
8: Return $\sigma, \{c_n\}_{n\in[N]}$

---

**Algorithm 6** DP-FedAvg-M with Adaptive Clipping (Andrew et al., 2021)

**Inputs:** No. clients $N$, No. global rounds $T$, No. local iterations $L$, noise multiplier $\sigma$, clip norm $c$, sampling rate $q$, loss functions $\{\mathcal{L}_n\}_{n\in[N]}$, local datasets $\{\mathcal{D}_n\}_{n\in[N]}$, client-side learning rate $\lambda$, server-side learning rate $\lambda_\text{s}$, clip-related learning rate $\lambda_\text{b}$, $\gamma$ quantile, batch size $B$, probability of violating $\delta$,

1: $\sigma =\text{SetSigma}(q, \epsilon, T, \delta, \sigma_\text{b})$
2: **Initialize** global model $\theta^0$
3: **for** each global round $t \in [T]$ **do**
4: $\quad \mathcal{C}^t \leftarrow$ Sample $qN$ clients uniformly.
5: $\quad$ **for** each client $n \in \mathcal{C}^t$ in parallel **do**
6: $\quad\quad (b_n^t, \tilde{\Delta}\theta_n^t) = \text{ClientUpdate}(t, n, \theta^{t-1}, c^t)$.
7: $\quad$ **end for**
8: $\quad$ Aggregate $\tilde{\Delta}\theta^t = \sum_{n\in\mathcal{C}^t} \tilde{\Delta}\theta_n^t$
9: $\quad$ Add noise $\tilde{\Delta}\theta^t \leftarrow \tilde{\Delta}\theta^t + \mathcal{N}(0, (c^t)^2\sigma^2\mathbb{I})$
10: $\quad$ Average $\tilde{\Delta}\theta^t \leftarrow \frac{1}{qN}\tilde{\Delta}\theta^t$
11: $\quad$ Compute $\tilde{\Delta}\theta^t \leftarrow \beta_\text{s}\tilde{\Delta}\theta^{t-1} + (1-\beta_\text{s})\tilde{\Delta}\theta^t$
12: $\quad$ Update $\theta^t = \theta^{t-1} + \lambda_\text{s}\tilde{\Delta}\theta^t$
13: $\quad c^{t+1}=\text{SetClippig}(\{b_n^t\}_{n\in\mathcal{C}^t}, \sigma_\text{b}, q, \gamma, \lambda_\text{b}, c^t)$
14: **end for**

**Def** `ClientUpdate`$(t, n, \theta^t, c^t)$

1: **Initialize** local model $\theta_n^{t,0} = \theta^t$
2: **for** local iteration $l \in [l]$ **do**
3: $\quad \{\mathcal{B}_i\}_{i=1}^{|\mathcal{D}_n|/B} \leftarrow$ Split $\mathcal{D}_n$ to size $B$ batches
4: $\quad$ **for** each batch $\mathcal{B}_i$ **do**
5: $\quad\quad \theta_n^{t,l} = \theta_n^{t,l-1} - \frac{\lambda \sum_{(\mathbf{x},y)\in\mathcal{B}_i} \nabla\mathcal{L}_n(\theta_n^{t,l-1};(\mathbf{x},y))}{B}$
6: $\quad$ **end for**
7: **end for**
8: Compute $\Delta\theta_n^t = \theta_n^{t,L} - \theta_n^{t,0}$
9: Compute $b = \mathcal{I}_{\|\Delta\theta_n^t\|\leq c^t}$
10: Clip $\tilde{\Delta}\theta_n^t = \Delta\theta_n^t \min\left(1, \frac{c^t}{\|\Delta\theta_n^t\|_2}\right)$
11: Return $b, \tilde{\Delta}\theta_n^t$

**Def** `SetSigma`$(q, \epsilon, T, \delta, \sigma_\text{b})$

1: $\bar{\sigma} =\text{GetNoise}(\epsilon, \delta, q, T)$
2: $\sigma = \left(\frac{1}{\bar{\sigma}^2} - \frac{1}{(2\sigma_\text{b})^2}\right)^{-1/2}$
3: Return $\sigma$

**Def** `SetClippig`$(\{b_n^t\}_{n\in\mathcal{C}^t}, \sigma_\text{b}, q, \gamma, \lambda_\text{b}, c^t)$

1: Aggregate $\bar{b}^t = \sum_{n\in\mathcal{C}^t} b_n^t$
2: Add noise $\bar{b}^t \leftarrow \bar{b}^t + \mathcal{N}(0, \sigma_\text{b}^2\mathbb{I})$
3: Average $\bar{b}^t \leftarrow \frac{1}{qN}\bar{b}^t$
4: Update $c^{t+1} = c^t \exp\left(-\lambda_\text{b}(\bar{b}^t - \gamma)\right)$
5: Return $c^{t+1}$

## A.2 PROOF OF LEMMA 1

We use induction to solve the recursive formula (1). According to (1), when $t = 1 < T_n$, $\epsilon_{\text{rdp},n}^1 = \frac{\epsilon_{\text{rdp},n}(q_n)^2}{T(q)^2}$, and when $t = 2 < T_n$, client $n$ spends $\epsilon_{\text{rdp},n}^2 = \frac{\epsilon_{\text{rdp},n} - \epsilon_{\text{rdp},n}^1}{T-1}\left(\frac{q_n}{q}\right)^2$. By substituting $\epsilon_{\text{rdp},n}^1$ in $\epsilon_{\text{rdp},n}^2$, we obtain $\epsilon_{\text{rdp},n}^2 = \frac{\epsilon_{\text{rdp},n}}{T-1}\left(1 - \frac{1}{T}\left(\frac{q_n}{q}\right)^2\right)\left(\frac{q_n}{q}\right)^2$. We now assume $\epsilon_{\text{rdp},n}^{t-1}$ satisfies in (2) for every $2 \le t < T$. If $t < T_n$, by substituting $\epsilon_{\text{rdp},n}^{t-1}$ in (1), we obtain

$$\epsilon_{\text{rdp},n}^t = \left(\frac{\epsilon_{\text{rdp},n} - \sum_{\tau=1}^{t-1}\epsilon_{\text{rdp},n}^\tau}{T-t+1}\right)\left(\frac{q_n}{q}\right)^2 = \left(\frac{\epsilon_{\text{rdp},n}^{t-1}(T-t+2)\left(\frac{q}{q_n}\right)^2 - \epsilon_{\text{rdp},n}^{t-1}}{T-t+1}\right)\left(\frac{q_n}{q}\right)^2 \tag{9}$$

$$= \epsilon_{\text{rdp},n}^{t-1}\frac{\left(T-t+2-\left(\frac{q_n}{q}\right)^2\right)}{T-t+1} \tag{10}$$

$$= \frac{\epsilon_{\text{rdp},n}}{T-t+2}\left(\frac{q_n}{q}\right)^2\left(\prod_{i=1}^{t-2}\left(1 - \frac{1}{T-t+2+i}\left(\frac{q_n}{q}\right)^2\right)\right)\frac{\left(T-t+2-\left(\frac{q_n}{q}\right)^2\right)}{T-t+1} \tag{11}$$

$$= \frac{\epsilon_{\text{rdp},n}}{T-t+1}\left(\frac{q_n}{q}\right)^2\prod_{i=1}^{t-1}\left(1 - \frac{1}{T-t+1+i}\left(\frac{q_n}{q}\right)^2\right). \tag{12}$$

If $t = T_n$, by substituting $\epsilon_{\text{rdp},n}^{t-1}$ in (1), we obtain

$$\epsilon_{\text{rdp},n}^{T_n} = \left(\frac{\epsilon_{\text{rdp},n} - \sum_{\tau=1}^{T_n-1}\epsilon_{\text{rdp},n}^\tau}{T-T_n+1}\right) = \left(\frac{\epsilon_{\text{rdp},n}^{T_n-1}(T-T_n+2)\left(\frac{q}{q_n}\right)^2 - \epsilon_{\text{rdp},n}^{T_n-1}}{T-T_n+1}\right) \tag{13}$$

$$= \epsilon_{\text{rdp},n}^{T_n-1}\frac{\left(T-T_n+2-\left(\frac{q_n}{q}\right)^2\right)}{T-T_n+1}\left(\frac{q}{q_n}\right)^2 \tag{14}$$

$$= \frac{\epsilon_{\text{rdp},n}}{T-T_n+2}\left(\prod_{i=1}^{T_n-2}\left(1 - \frac{1}{T-T_n+2+i}\left(\frac{q_n}{q}\right)^2\right)\right)\frac{\left(T-T_n+2-\left(\frac{q_n}{q}\right)^2\right)}{T-T_n+1} \tag{15}$$

$$= \frac{\epsilon_{\text{rdp},n}}{T-T_n+1}\prod_{i=1}^{T_n-1}\left(1 - \frac{1}{T-T_n+1+i}\left(\frac{q_n}{q}\right)^2\right). \tag{16}$$

If $t > T_n$, by substituting $\epsilon_{\text{rdp},n}^{t-1}$ in (1), we obtain

$$\epsilon_{\text{rdp},n}^t = \left(\frac{\epsilon_{\text{rdp},n} - \sum_{\tau=1}^{t-1}\epsilon_{\text{rdp},n}^\tau}{T-t+1}\right) = \left(\frac{\epsilon_{\text{rdp},n}^{t-1}(T-t+2) - \epsilon_{\text{rdp},n}^{t-1}}{T-t+1}\right) \tag{17}$$

$$= \epsilon_{\text{rdp},n}^{t-1} = \frac{\epsilon_{\text{rdp},n}}{T-T_n+1}\prod_{i=1}^{T_n-1}\left(1 - \frac{1}{T-T_n+1+i}\left(\frac{q_n}{q}\right)^2\right). \tag{18}$$

## A.3 PROOF OF THEOREM 1

We use the explicit solutions of the recursive formula (1), presented in Lem. 1, to prove this theorem. When $t < T_n$,

$$\epsilon_{\text{rdp},n}^{t} - \epsilon_{\text{rdp},n}^{t-1} = \frac{\epsilon_{\text{rdp},n}}{T-t+1} \left(\frac{q_n}{q}\right)^2 \prod_{i=1}^{t-1} \left(1 - \frac{1}{T-t+1+i} \left(\frac{q_n}{q}\right)^2\right) \tag{19}$$

$$- \frac{\epsilon_{\text{rdp},n}}{T-t+2} \left(\frac{q_n}{q}\right)^2 \prod_{i=1}^{t-2} \left(1 - \frac{1}{T-t+2+i} \left(\frac{q_n}{q}\right)^2\right) \tag{20}$$

$$= \epsilon_{\text{rdp},n} \left(\frac{q_n}{q}\right)^2 \left(\prod_{i=1}^{t-2} \left(1 - \frac{1}{T-t+2+i} \left(\frac{q_n}{q}\right)^2\right)\right) \tag{21}$$

$$\times \left(\frac{1}{T-t+1} \left(1 - \frac{1}{T-t+2} \left(\frac{q_n}{q}\right)^2\right) - \frac{1}{T-t+2}\right) \tag{22}$$

$$= \epsilon_{\text{rdp},n} \left(\frac{q_n}{q}\right)^2 \left(\prod_{i=1}^{t-2} \left(1 - \frac{1}{T-t+2+i} \left(\frac{q_n}{q}\right)^2\right)\right) \frac{1 - \left(\frac{q_n}{q}\right)^2}{(T-t+1)(T-t+2)}. \tag{23}$$

The right-hand side of (23) is larger than equal to zero because $q_n \leq q$. Therefore, in this case $\epsilon_{\text{rdp},n}^{t} \geq \epsilon_{\text{rdp},n}^{t-1}$. When $t = T_n$,

$$\epsilon_{\text{rdp},n}^{T_n} - \epsilon_{\text{rdp},n}^{T_n-1} = \frac{\epsilon_{\text{rdp},n}}{T-T_n+1} \prod_{i=1}^{T_n-1} \left(1 - \frac{1}{T-T_n+1+i} \left(\frac{q_n}{q}\right)^2\right) \tag{24}$$

$$- \frac{\epsilon_{\text{rdp},n}}{T-T_n+2} \left(\frac{q_n}{q}\right)^2 \prod_{i=1}^{T_n-2} \left(1 - \frac{1}{T-T_n+2+i} \left(\frac{q_n}{q}\right)^2\right) \tag{25}$$

$$= \epsilon_{\text{rdp},n} \left(\prod_{i=1}^{T_n-2} \left(1 - \frac{1}{T-T_n+2+i} \left(\frac{q_n}{q}\right)^2\right)\right) \tag{26}$$

$$\times \left(\frac{1}{T-t+1} \left(1 - \frac{1}{T-T_n+2} \left(\frac{q_n}{q}\right)^2\right) - \frac{\left(\frac{q_n}{q}\right)^2}{T-T_n+2}\right) \tag{27}$$

$$= \epsilon_{\text{rdp},n} \left(\prod_{i=1}^{T_n-2} \left(1 - \frac{1}{T-T_n+2+i} \left(\frac{q_n}{q}\right)^2\right)\right) \frac{1 - \left(\frac{q_n}{q}\right)^2}{(T-T_n+1)}. \tag{28}$$

The right-hand side of (28) is again larger than equal to zero because. Therefore, in this case we also have $\epsilon_{\text{rdp},n}^{T_n} \geq \epsilon_{\text{rdp},n}^{T_n-1}$. Lem. 1 also shows $\epsilon_{\text{rdp},n}^{t} = \epsilon_{\text{rdp},n}^{t-1}$ when $t > T_n$.

## A.4 Proof of Theorem 2

If the expectation is taken w.r.t. (i) the randomness of local datasets, (ii) the sampling of clients, and (iii) the randomness of injected Gaussian noise, then the bias is simplified as follows:

$$\left\| \mathbb{E}_{(i),(ii),(iii)} \left( \text{Error}^t \right) \right\|$$

$$= \frac{1}{N} \left\| \sum_{n=1}^{N} \mathbb{E}_{(i),(ii),(iii)} \left( \Delta\theta_n^t - \frac{1}{q^t} \left( \mathbf{b}_n^t \left( \text{Clip} \left( \Delta\theta_n^t, c_n^t \right) + \mathbf{z}_n^t \right) + (1 - \mathbf{b}_n^t) \tilde{\mathbf{z}}_n^t \right) \right) \right\|$$

$$= \frac{1}{N} \left\| \sum_{n=1}^{N} \mathbb{E}_{(i),(ii)} \left( \Delta\theta_n^t - \frac{1}{q^t} \mathbf{b}_n^t \left( \text{Clip} \left( \Delta\theta_n^t, c_n^t \right) \right) \right) \right\| \tag{29}$$

$$= \frac{1}{N} \left\| \sum_{n=1}^{N} \mathbb{E}_{(i)} \left( \Delta\theta_n^t - \frac{q_n^t}{q^t} \left( \text{Clip} \left( \Delta\theta_n^t, c_n^t \right) \right) \right) \right\| \tag{30}$$

$$= \frac{1}{N} \left\| \sum_{n=1}^{N} \mathbb{E}_{(i)} \left( \Delta\theta_n^t \left( \frac{q^t}{q^t} - \frac{q_n^t}{q^t} + \frac{q_n^t}{q^t} \right) - \frac{q_n^t}{q^t} \left( \text{Clip} \left( \Delta\theta_n^t, c_n^t \right) \right) \right) \right\| \tag{31}$$

$$= \frac{1}{N} \left\| \sum_{n=1}^{N} \left( \left( \frac{q^t}{q^t} - \frac{q_n^t}{q^t} \right) \mathbb{E}_{(i)} \left( \Delta\theta_n^t \right) + \frac{q_n^t}{q^t} \mathbb{E}_{(i)} \left( \Delta\theta_n^t - \left( \text{Clip} \left( \Delta\theta_n^t, c_n^t \right) \right) \right) \right) \right\| \tag{32}$$

$$\leq \frac{1}{N} \left\| \sum_{n=1}^{N} \left( \frac{q^t}{q^t} - \frac{q_n^t}{q^t} \right) \mathbb{E}_{(i)} \left( \Delta\theta_n^t \right) \right\| + \frac{1}{N} \sum_{n=1}^{N} \left\| \frac{q_n^t}{q^t} \mathbb{E}_{(i)} \left( \Delta\theta_n^t - \left( \text{Clip} \left( \Delta\theta_n^t, c_n^t \right) \right) \right) \right\| \tag{33}$$

$$\leq \frac{1}{N} \left\| \sum_{n=1}^{N} \left( \frac{q^t}{q^t} - \frac{q_n^t}{q^t} \right) \mathbb{E}_{(i)} \left( \Delta\theta_n^t \right) \right\| + \frac{1}{N} \sum_{n=1}^{N} \frac{q_n^t}{q^t} \frac{\mathbb{E}_{(i)} \left( \| \Delta\theta_n^t \|^\rho \right)}{(c_n^t)^{\rho-1}}. \tag{34}$$

The equality (29) is due to $\mathbb{E}_{(iii)} \left( \mathbf{z}_n^t \right) = \mathbb{E}_{(iii)} \left( \tilde{\mathbf{z}}_n^t \right) = 0$. The equality (30) is due to $\mathbb{E}_{(ii)} \left( \mathbf{b}_n^t \right) = q_n^t$. The inequality (33) is due to triangle inequality. The inequality (34) is due to the clipping bias lemma (Das et al., 2023), given any $\rho > 1$.

## A.5 Proof of Theorem 3

If the expectation is taken w.r.t. (i) the randomness of local datasets and (ii) the sampling of clients, (iii) the randomness of injected Gaussian noise, and (iv) privacy budget assignment randomness, then the bias is simplified as follows:

$$\left\| \mathbb{E}_{(i),(ii),(iii),(iv)} \left( \text{Error}^t \right) \right\|$$

$$= \frac{1}{N} \left\| \sum_{n=1}^{N} \mathbb{E}_{(i),(ii),(iii),(iv)} \left( \Delta\theta_n^t - \frac{1}{q^t} \left( \mathsf{b}_n^t \left( \text{Clip} \left( \Delta\theta_n^t, c_n^t \right) + z_n^t \right) + (1 - \mathsf{b}_n^t) \tilde{z}_n^t \right) \right) \right\|$$

$$= \frac{1}{N} \left\| \sum_{n=1}^{N} \mathbb{E}_{(i),(ii),(iv)} \left( \Delta\theta_n^t - \frac{1}{q^t} \mathsf{b}_n^t \left( \text{Clip} \left( \Delta\theta_n^t, c_n^t \right) \right) \right) \right\| \tag{35}$$

$$= \frac{1}{N} \left\| \sum_{n=1}^{N} \mathbb{E}_{(i),(ii),(iv)} \left( \Delta\theta_n^t - \frac{\mathsf{b}_n^t}{q^t} \left( \text{Clip} \left( \Delta\theta_n^t, c_n^t \right) \right) \right) \right\| \tag{36}$$

$$= \frac{1}{N} \left\| \sum_{n=1}^{N} \mathbb{E}_{(i),(iv)} \left( \Delta\theta_n^t - \frac{q_n^t}{q^t} \left( \text{Clip} \left( \Delta\theta_n^t, c_n^t \right) \right) \right) \right\| \tag{37}$$

$$= \frac{1}{N} \left\| \sum_{n=1}^{N} \mathbb{E}_{(i),(iv)} \left( \frac{\Delta\theta_n^t}{1} \left( \frac{q^t}{q^t} - \frac{q_n^t}{q^t} + \frac{q_n^t}{q^t} \right) - \frac{q_n^t}{q^t} \left( \text{Clip} \left( \Delta\theta_n^t, c_n^t \right) \right) \right) \right\| \tag{38}$$

$$= \frac{1}{N} \left\| \sum_{n=1}^{N} \left( \mathbb{E}_{(i),(iv)} \left( \left( \frac{q^t}{q^t} - \frac{q_n^t}{q^t} \right) \Delta\theta_n^t \right) + \mathbb{E}_{(i),(iv)} \left( q_n^t q^t \Delta\theta_n^t - \left( \text{Clip} \left( \Delta\theta_n^t, c_n^t \right) \right) \right) \right) \right\| \tag{39}$$

$$\leq \frac{1}{N} \left\| \sum_{n=1}^{N} \mathbb{E}_{(iv)} \left( \left( \frac{q^t}{q^t} - \frac{q_n^t}{q^t} \right) \mathbb{E}_{(i)} \left( \Delta\theta_n^t \right) \right) \right\| +$$

$$+ \frac{1}{N} \sum_{n=1}^{N} \left\| \mathbb{E}_{(iv)} \left( \frac{q_n^t}{q^t} \mathbb{E}_{(i)} \left( \Delta\theta_n^t - \left( \text{Clip} \left( \Delta\theta_n^t, c_n^t \right) \right) \right) \right) \right\| \tag{40}$$

$$\leq \frac{1}{N} \left\| \sum_{n=1}^{N} \mathbb{E}_{(iv)} \left( \left( \frac{q^t}{q^t} - \frac{q_n^t}{q^t} \right) \mathbb{E}_{(i)} \left( \Delta\theta_n^t \right) \right) \right\| +$$

$$+ \frac{1}{N} \sum_{n=1}^{N} \mathbb{E}_{(iv)} \left( \left\| \frac{q_n^t}{q^t} \mathbb{E}_{(i)} \left( \Delta\theta_n^t - \left( \text{Clip} \left( \Delta\theta_n^t, c_n^t \right) \right) \right) \right\| \right) \tag{41}$$

$$\leq \frac{1}{N} \left\| \sum_{n=1}^{N} \mathbb{E}_{(iv)} \left( \left( \frac{q^t}{q^t} - \frac{q_n^t}{q^t} \right) \mathbb{E}_{(i)} \left( \Delta\theta_n^t \right) \right) \right\| + \frac{1}{N} \sum_{n=1}^{N} \mathbb{E}_{(iv)} \left( \frac{q_n^t}{q^t} \frac{\mathbb{E}_{(i)} \left( \|\Delta\theta_n^t\|^\rho \right)}{(c_n^t)^{\rho-1}} \right). \tag{42}$$

The equality (35) is due to $\mathbb{E}_{(iii)} \left( z_n^t \right) = \mathbb{E}_{(iii)} \left( \tilde{z}_n^t \right) = 0$. The equality (36) is due to $\mathbb{E}_{(ii)} \left( \mathsf{b}_n^t \right) = q_n^t$. The inequality (40) is due to triangle inequality. The inequality (42) is due to the clipping bias lemma (Das et al., 2023) given any $\rho > 1$.

We next further simplify the first and second terms on the right-hand side of (42).

The first term equals zero:

$$\mathbb{E}_{(iv)}\left(\left(\frac{q^t}{q^t}-\frac{q^t_n}{q^t}\right)\mathbb{E}_{(i)}\left(\Delta\theta^t_n\right)\right)$$

$$= \mathbb{E}_{(i)}\left(\left\|\Delta\theta^t_n\right\|^\rho\right)\mathbb{E}_{(iv)}\left(1-\frac{q^t_n}{q^t}\right) \tag{43}$$

$$= \mathbb{E}_{(i)}\left(\left\|\Delta\theta^t_n\right\|^\rho\right)\left(\sum_{i\in[N]}\Pr\left(\epsilon_n=\hat{\epsilon}_i\right)\mathbb{E}_{(iv)}\left(1-\frac{q^t_n}{q^t}\middle|\epsilon_n=\hat{\epsilon}_i\right)\right) \tag{44}$$

$$= \mathbb{E}_{(i)}\left(\left\|\Delta\theta^t_n\right\|^\rho\right)\left(\frac{1}{N}\sum_{i\in[N]}\left(1-\frac{\hat{q}^t_i}{q^t}\right)\right) \tag{45}$$

$$= \mathbb{E}_{(i)}\left(\left\|\Delta\theta^t_n\right\|^\rho\right)\left(\frac{1}{N}\sum_{n\in[N]}\left(1-\frac{q^t_n}{q^t}\right)\right) \tag{46}$$

$$= \mathbb{E}_{(i)}\left(\left\|\Delta\theta^t_n\right\|^\rho\right)\left(\frac{1}{N}\left(N-\frac{q^tN}{q^t}\right)\right)=0. \tag{47}$$

Equality (43) is due to the independency of randomness between (i) and (iv). Equality (45) is because of our assumption that the sampling from $P_n$ and of $\epsilon_n$ is independent. Equality (47) is because $\sum_{n=1}^N q^t_n = q^t$.

The second term on the right-hand side of (42) can be further simplified into:

$$\mathbb{E}_{(iv)}\left(\frac{q^t_n}{q^t}\frac{\mathbb{E}_{(i)}\left(\left\|\Delta\theta^t_n\right\|^\rho\right)}{(c^t_n)^{\rho-1}}\right)$$

$$= \mathbb{E}_{(i)}\left(\left\|\Delta\theta^t_n\right\|^\rho\right)\mathbb{E}_{(iv)}\left(\frac{q^t_n}{q^t}\frac{1}{(c^t_n)^{\rho-1}}\right) \tag{48}$$

$$= \mathbb{E}_{(i)}\left(\left\|\Delta\theta^t_n\right\|^\rho\right)\left(\sum_{i\in[N]}\Pr\left(\epsilon_n=\hat{\epsilon}_i\right)\mathbb{E}_{(iv)}\left(\frac{q^t_n}{q^t}\frac{1}{(c^t_n)^{\rho-1}}\middle|\epsilon_n=\hat{\epsilon}_i\right)\right) \tag{49}$$

$$= \mathbb{E}_{(i)}\left(\left\|\Delta\theta^t_n\right\|^\rho\right)\left(\frac{1}{N}\sum_{i\in[N]}\frac{\hat{q}^t_i}{q^t}\frac{1}{(\hat{c}^t_i)^{\rho-1}}\right) \tag{50}$$

$$= \mathbb{E}_{(i)}\left(\left\|\Delta\theta^t_n\right\|^\rho\right)\left(\frac{1}{N}\sum_{n\in[N]}\frac{q^t_n}{q^t}\frac{1}{(c^t_n)^{\rho-1}}\right). \tag{51}$$

Equality (48) is due to the independency of randomness between (i) and (iv). Equality (49) is because of our assumption that the sampling from $P_n$ and of $\epsilon_n$ is independent. Combining (42), (47), and (51), Thm. 3 is proved.

## A.6 EXTENDED BACKGROUND

### A.6.1 SOME USEFUL LEMMAS FROM PRIOR WORKS

**Lemma 2. [Clipping bias (Das et al., 2023)** *Suppose $\phi(\xi)$ (where $\xi$ denotes the source of randomness) is an unbiased estimator of $\phi$, i.e., $\mathbb{E}_\xi\left(\phi(\xi)\right)=\phi$. Let $b(\xi)$ denote the clipping bias of $Clip\left(\phi(\xi),c\right)$, i.e., $b(c)=\left\|\phi-\mathbb{E}_\xi\left(Clip\left(\phi(\xi),c\right)\right)\right\|$. Then for any $\rho>1$,*

$$b(c)\le\frac{\mathbb{E}_\xi\left(\left\|\phi(\xi)\right\|^\rho\right)}{c^{\rho-1}}. \tag{52}$$

### A.6.2 DIFFERENTIAL PRIVACY

**Definition 1** (($\epsilon, \delta$)-DP Dwork et al. (2014)). *The randomized algorithm $\mathcal{A} : \chi \to \mathcal{R}$ with domain $\chi$ and range $\mathcal{R}$ satisfies ($\epsilon, \delta$)-DP iff for any two* neighboring *inputs $\mathcal{D}, \mathcal{D}' \in \chi$ that differ by at most one record, and any measurable subset of outputs $\mathcal{S} \subseteq \mathcal{R}$,*

$$Pr\left(\mathcal{A}(\mathcal{D}) \in \mathcal{S}\right) \le e^\epsilon Pr\left(\mathcal{A}(\mathcal{D}') \in \mathcal{S}\right) + \delta. \tag{53}$$

In (53), the privacy budget $\epsilon \in \mathbb{R}_+$ controls the extent to which the output distributions induced by two neighboring inputs may differ. The $\delta \in [0, 1]$ quantifies the probability of violating the privacy guarantee. Allowing a larger $\delta \in [0, 1]$ improves utility at the cost of a more relaxed (weaker) privacy guarantee. One way to relax the DP guarantee is to use ($\alpha, \epsilon$)-Rényi DP (RDP) (Mironov, 2017). The $\alpha > 1$ is the order of Rényi divergence between distributions $P := \Pr\left(\mathcal{A}(\mathcal{D})\right)$ and $P' := \Pr\left(\mathcal{A}(\mathcal{D}')\right)$, defined as

$$\boldsymbol{R}_\alpha\left(P \| P'\right) := \frac{1}{1 - \alpha} \log \mathbb{E}_{x \sim P'} \left(\frac{P}{P'}\right)^\alpha. \tag{54}$$

While the Rényi divergence can be defined for $\alpha < 1$, including negative orders, the RDP definition Mironov (2017) is based on $\alpha \ge 1$ and is outlined as follows.

**Definition 2** (Rényi DP (RDP) Mironov (2017)). *The randomized algorithm $\mathcal{A} : \chi \to \mathcal{R}$ with domain $\chi$ and range $\mathcal{R}$ is ($\alpha, \epsilon$)-RDP iff for any neighboring inputs $\mathcal{D}, \mathcal{D}' \in \chi$, we have*

$$\boldsymbol{R}_\alpha\left(Pr\left(\mathcal{A}(\mathcal{D})\right) \| Pr\left(\mathcal{A}(\mathcal{D}')\right)\right) \le \epsilon. \tag{55}$$

When accounting for total privacy consumption over an iterative algorithm, RDP offers a smoother composition property than DP. RDP allows the privacy budget to accumulate linearly with the number of training rounds (Mironov, 2017). This simplifies the tracking and management of privacy budgets over time. We next recall a lemma from (Mironov, 2017). Lemma 3 shows how RDP can be converted to DP when needed.

**Lemma 3.** *If $\mathcal{A}$ is an ($\alpha, \epsilon_{rdp}$)-RDP algorithm, it also satisfies ($\epsilon, \delta$)-DP for any $0 < \delta < 1$, where*

$$\epsilon = \epsilon_{rdp} + \log \frac{\alpha - 1}{\alpha} - \frac{\log \delta + \log \alpha}{\alpha - 1}. \tag{56}$$

To implement privacy guarantees, we use the sampled Gaussian mechanism (SGM) Mironov et al. (2019), formally defined as follows.

**Definition 3** (SGM Mironov et al. (2019)). *Consider the algorithm $\mathcal{A}$ which maps a subset $\mathcal{D} \subseteq \chi$ to $\mathbb{R}^w$ and has $\ell_2$-sensitivity $c$. The sampled Gaussian mechanism parameterized by the sampling rate $q \in [0, 1]$, $c$, and noise multiplier $\sigma > 0$ is defined as*

$$\mathcal{G}_{\sigma,c,q}(\mathcal{D}) := \mathcal{A}(\{x \mid x \in \mathcal{D} \text{ is sampled with Probability } q\}) + \mathcal{N}(0, c^2\sigma^2\mathbb{I}_w), \tag{57}$$

*where each element of $\mathcal{D}$ is (Poisson) sampled independently at random with probability $q$, and $\mathcal{N}(0, c^2\sigma^2\mathbb{I}_w)$ is spherical $w$-dimensional Gaussian noise with per-coordinate variance $c^2\sigma^2$.*

**Lemma 4** (Mironov et al. (2019)). *The SGM $\mathcal{G}_{\sigma,c,q}$ with $c = 1$ guarantees ($\alpha, \epsilon$)-RDP, where $\epsilon \le \frac{2\alpha q^2}{\sigma^2}$.*

### A.7 EXTENDED EXPERIMENTAL SETUP

We conduct our experiments in Python 3.11 using Pytorch leveraging the $4 \times$ L4 24 GB GPU. Below, we provide additional details on the experimental setups used in Sec. 5 to analyze how our time-adaptive DP-FL framework enhances the privacy-utility tradeoff and in Appendix A.8 which extends experiments for further analysis.

**Details on Datasets.** For our experiments, we use FMNIST, MNIST, Adult Income, and CIFAR10 datasets. Both FMNIST and MNIST datasets have a training set of 60,000 and a test set of 10,000 $28 \times 28$ images, associated with 10 labels. The Adult Income dataset consists of 48,842 samples with 14 features and is split into a training set of 32,561 samples and a test set of 16,281 samples.

The CIFAR10 dataset consists of 60,000 $32 \times 32$ color images in 10 classes, with 6000 images per class. There are 50,000 training images and 10,000 test images.

**Simulation Parameters.** Throughout our simulations, we use SGD optimizer and momentum equal to 0.9. We also use a CosineAnnealing learning rate scheduler from (Loshchilov & Hutter, 2016) for faster convergence. In Sec.5, we fix the spending-based sample rate (during spend mode) to $q = 0.9$ and the average clipping norm to $c = 250$. We consider the transition from saving round to spending round occurs in the middle of training. I.e., given the total number of rounds $T = 25$, we set $T_{\text{group},1} = T_{\text{group},2}, T_{\text{group},3} = 13$. The obtained results are averaged over three runs. In Table 4 we summarize other hyperparameters, including learning rate ($\lambda$), number of clients ($N$), batch size ($B$), number of local epochs ($L$), and the saving-based sampling rates of clients from privacy groups 1, 2, and 3 ($q_{\text{group},1}, q_{\text{group},2}, q_{\text{group},3}$).

Table 4: Parameters for different datasets, used in Table 2 and Figure 2. We set $T = 25$, $T_{\text{group},1} = T_{\text{group},2}, T_{\text{group},3} = 13$, $q = 0.9$, and $c = 250$.

| **Dataset** | $(\epsilon_{\text{group},1}, \epsilon_{\text{group},2}, \epsilon_{\text{group},3})$ | $\lambda$ | $N$ | $B$ | $L$ | $(q_{\text{group},1}, q_{\text{group},2}, q_{\text{group},3})$ |
|---|---|---|---|---|---|---|
| FMNIST | $(10, 20, 30)$ | 0.001 | 100 | 125 | 30 | $(0.5, 0.6, 0.7)$ |
| MNIST | $(10, 15, 20)$ | 0.001 | 100 | 125 | 30 | $(0.5, 0.6, 0.7)$ |
| Adult Income | $(10, 20, 30)$ | 0.01 | 80 | 32 | 5 | $(0.6, 0.7, 0.8)$ |

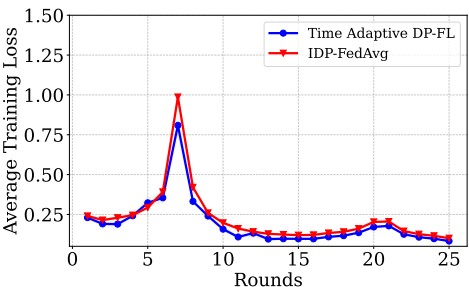

Figure 3: Average Training loss of clients in our time-adaptive DP-FL scheme plotted versus the IDP-FedAvg baseline with FMNIST dataset in training rounds T = 25. We set $(\epsilon_{\text{group},1}, \epsilon_{\text{group},2}, \epsilon_{\text{group},3}) = (10, 20, 30)$ in our scheme and IDP-FedAvg.

## A.8 EXTENDED EXPERIMENTAL RESULTS

**Impact of Training Rounds on Model Convergence.** We extend experiments to more training rounds— $T \in \{25, 50, 100\}$. For example, in Figure 4, we set $T = 50$, and plot the global test accuracy vs. communication rounds. It is evident from Figure 4 that for our time-adaptive DP-FL framework, as the number of training rounds increases, the upward trend in the accuracy starts slowing down. However, increasing the number of communication rounds does not always improve accuracy. This is because, with more rounds, the privacy budget is distributed across more rounds, resulting in a lower budget per round. Consequently, the increased effect of perturbation can degrade the privacy-utility tradeoff. This is demonstrated in our FMNIST and MNIST experiments, as shown in Table 5, in which we report the final-round test accuracy across different schemes. As shown in the third column of Table 5, when training rounds increase from 25 to 50 and from 50 to 100, FedAvg (the non-DP baseline scheme) consistently demonstrates an upward trend in both MNIST and FMNIST experiments. However, our scheme (fifth column of Table 5) and IDP-FedAvg (fourth column), which operate under limited group privacy budgets $(\epsilon_{\text{group},1}, \epsilon_{\text{group},2}, \epsilon_{\text{group},3}) = (10, 20, 30)$, do not exhibit the same consistent improvement. They exhibit an upward trend from 25 to 50 rounds but not consistently from 50 to 100 rounds. Notably, the best performance amongst the DP experiments of Table 5) is achieved by our scheme, reaching 75.63% after $T = 100$ rounds for the FMNIST dataset, and 90.78% at $T = 50$ rounds for the MNIST dataset.

**Impact of Different Privacy Budgets on Model Utility.** We present additional experimental results to evaluate the impact of stricter privacy budgets $(\epsilon_{\text{group},1}, \epsilon_{\text{group},2}, \epsilon_{\text{group},3}) = (2, 5, 10)$ and $(5, 5, 5)$

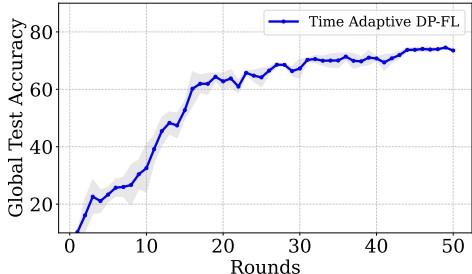

Figure 4: **Global test accuracy for increasing number of communication rounds.** In this figure, we use the FMNIST dataset, $N = 100$ clients, $L = 30$ local iterations, $(\epsilon_{\text{group},1}, \epsilon_{\text{group},2}, \epsilon_{\text{group},3}) = (20, 20, 20)$, $c = 250$, and $q = 0.8$.

Table 5: Benchmarking our time-adaptive DP-FL scheme against the baselines in terms of global test accuracy and across varying datasets and number of training rounds (T). We set $(\epsilon_{\text{group},1}, \epsilon_{\text{group},2}, \epsilon_{\text{group},3}) = (10, 20, 30)$ in our scheme and IDP-FedAvg.

| Dataset | T | FedAvg (non-DP) | IDP-FedAvg | Ours |
|---------|-----|---------|------------|------|
| FMNIST | 25 | 72.95 | 62.57 | **66.55** |
| | 50 | 76.00 | 71.80 | **75.51** |
| | 100 | 80.14 | 71.29 | **75.63** |
| MNIST | 25 | 90.23 | 64.53 | **74.69** |
| | 50 | 93.42 | 89.57 | **90.78** |
| | 100 | 95.91 | 87.00 | **90.15** |

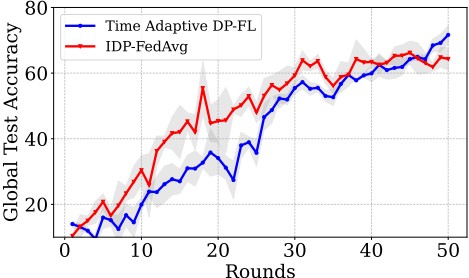

Figure 5: **Test accuracy for our time-adaptive DP-FL framework vs. IDP-FedAvg, using stricter privacy budgets** $(\epsilon_{\textbf{group},1}, \epsilon_{\textbf{group},2}, \epsilon_{\textbf{group},3}) = (2, 5, 10)$**.** In this figure, we use $N = 100$ clients, $T = 50$ global rounds, $L = 30$ local iterations, $c = 250$, and $q = 0.8$.

on model utility (test accuracy). The results are presented in Figure 5 and Tables 6 and 7. As expected, we observe that lower privacy budgets hamper utility. In particular, in Table 6, we benchmark our scheme against the IDP-FedAvg baseline using two sets of non-uniform privacy budgets, $(10, 20, 30)$ and $(2, 5, 10)$, evaluated across two datasets. Our findings suggest that the time-adaptive DP-FL framework yields considerably higher utility than IDP-FedAvg, also under stringent privacy constraints. Similarly, Table 7 focuses on uniform privacy budgets and further confirms that even with a reduction in privacy budgets from $(10, 10, 10)$ to $(5, 5, 5)$, our scheme consistently outperforms the corresponding baselines.

**Additional Baseline.** We benchmark our scheme against the adaptive clipping method Andrew et al. (2021), with pseudocode provided in Algorithm 6. We present results in the third and fifth columns of Table 7. This baseline is designed for uniform privacy budgets and is parameterized by the server-side learning rate $\lambda_{\text{s}}$ and momentum parameter $\beta_{\text{s}}$, which are not privacy-specific. To ensure a fair comparison with our scheme and other baselines in our paper, we set these parameters to $\lambda_{\text{s}} = 1.0$

Table 6: Benchmarking our time-adaptive DP-FL scheme against the baselines in terms of the final-round test accuracy and across varying privacy budgets $(\epsilon_{\text{group},1}, \epsilon_{\text{group},2}, \epsilon_{\text{group},3})$. We set $T = 25$ and $L = 30$ for $\epsilon = \{10, 20, 30\}$ and $T = 25$ and $L = 50$ for $\epsilon = \{2, 5, 10\}$, $(q_{\text{group},1}, q_{\text{group},2}, q_{\text{group},3}) = (0.3, 0.5, 0.7)$.

| Dataset | Privacy Budgets | IDP-FedAvg Non-uniform | Ours Non-uniform |
|---|---|---|---|
| FMNIST | $(10, 20, 30)$ | 62.57 | **66.55** |
| FMNIST | $(2, 5, 10)$ | 60.99 | **65.75** |
| MNIST | $(10, 20, 30)$ | 64.53 | **77.38** |
| MNIST | $(2, 5, 10)$ | 63.35 | **66.50** |

Table 7: Benchmarking our time-adaptive DP-FL scheme against the baselines in terms of the final-round test accuracy and across varying uniform privacy budgets $\epsilon_{\text{group},1} = \epsilon_{\text{group},2} = \epsilon_{\text{group},3}$. We set $T = 25$ and $L = 30$.

| Dataset | Privacy Budgets | Adaptive Clipping $(\beta_s, \lambda_s) = (0, 1.0)$ | DP-FedAvg | Adaptive Clipping optimal $(\beta_s, \lambda_s)$ | Ours |
|---|---|---|---|---|---|
| FMNIST | $(10, 10, 10)$ | 60.23 | 64.8 | 67.64 | **67.90** |
| FMNIST | $(5, 5, 5)$ | 52.39 | 51.06 | 52.39 | **60.79** |
| MNIST | $(10, 10, 10)$ | 65.59 | 76.79 | 78.04 | **80.2** |
| MNIST | $(5, 5, 5)$ | 55.48 | 61.45 | 55.48 | **69.07** |

and $\beta_s = 0.0$. In column 3, we use these default values, while in column 5, we select the optimal values from a set of possible choices. As shown in the table, our scheme consistently outperforms adaptive clipping, even when the baseline's parameters are optimally tuned.

**Effect of Number of Clients on Model Utility.** We experiment with different numbers of clients—$N \in \{30, 60, 75\}$—for the MNIST dataset to validate the applicability of our time-adaptive DP-FL framework across various scenarios. Additionally, we also perform experiments to analyze if our framework outperforms the baselines, in terms of the utility of the trained model. Our results in Table 8 indicate that for all the different numbers of clients that we consider, our framework remarkably surpasses the utility of the baseline.

Table 8: Comparison of model utility on a varying number of clients and comparison of model utility for time-adaptive DP-FL with baselines for a varying number of clients

| Number of clients | SETUP Privacy Budgets | IDP-FedAvg Non-uniform | Ours Non-uniform |
|---|---|---|---|
| 30 | $(10, 20, 30)$ | 72.35 | **73.34** |
| 60 | $(10, 20, 30)$ | 78.69 | **83.83** |
| 75 | $(10, 20, 30)$ | 70.93 | **77.53** |

**The Choice of Hyperparameters.** We evaluate our DP-FL framework with different choices of hyperparameters—different saving-based sampling rates $(q_{\text{group},1}, q_{\text{group},2}, q_{\text{group},3})$ and different saving-to-spending transition rounds $(T_{\text{group},1}, T_{\text{group},2}, T_{\text{group},3})$. The final-round test accuracies for different choices of $(q_{\text{group},1}, q_{\text{group},2}, q_{\text{group},3})$, and across both MNIST and FMNIST datasets, are presented in Table 9. In this table, in Column 3 we set these rates as $(0.5, 0.6, 0.7)$, in Column 4 as $(0.3, 0.5, 0.7)$, and in Column 5 as $(0.6, 0.6, 0.6)$. As shown in the table, our scheme, which uses lower sampling rates during saving—e.g., for all $i \in [3]$, $q_{\text{group},1}$ is smaller than $q = 0.9$ in this table—outperforms the IDP-FedAvg baseline (Column 6) that uses a uniform sampling rate $q$ over time. This table also shows that our method is relatively robust against the clients' choice of saving-based sampling rates, consistently achieving performance between that of IDP-FedAvg and the ideal case of FedAvg without DP (Column 2).

Table 9: Evaluating the impact of saving-based sampling rates of different privacy groups, $(q_{\text{group},1}, q_{\text{group},2}, q_{\text{group},3})$, on our time-adaptive DP-FL scheme in comparison with the baseline. We set $(\epsilon_{\text{group},1}, \epsilon_{\text{group},2}, \epsilon_{\text{group},3}) = (10, 20, 30)$, $T = 25$, $L = 30$ and $N = 100$, $T_{\text{group},1} = T_{\text{group},2} = T_{\text{group},3} = 13$, $q = 0.9$, $c = 250$, $\lambda = 0.001$, and $B = 125$.

| DATASET | FedAvg non-DP | Ours $(0.5, 0.6, 0.7)$ | Ours $(0.3, 0.5, 0.7)$ | Ours $(0.6, 0.6, 0.6)$ | IDP-FedAvg |
|---------|---------------|------------------------|------------------------|------------------------|------------|
| MNIST | 90.23 | **72.72** | **77.39** | **71.6** | 64.53 |
| FMNIST | 72.95 | **70.57** | **66.55** | **67.75** | 62.57 |

The final-round test accuracies for different choices of saving-to-spending transition rounds $(T_{\text{group},1}, T_{\text{group},2}, T_{\text{group},3})$, for both MNIST and FMNIST datasets, are presented in Table 10. We set the total number of rounds as $T = 25$. In this table, in Column 3 we set the transition rounds as $(7,7,7)$, in Column 4 as $(7, 13, 19)$, in Column 5 as $(19, 13, 7)$, and in Column 6 as $(19, 19, 19)$. As shown in the table, our scheme, which transitions from saving to spend mode sometime between the first and final round—i.e., for all $i \in [3]$, $1 < T_{\text{group},1} < 25$—outperforms the IDP-FedAvg baseline (Column 7) which can be viewed as a special case of ours with transition rounds set to $(1, 1, 1)$. This table shows the robustness of our method to the client's choice of transition rounds, showing less than a 2% variation in accuracy across different choices while consistently achieving performance between that of IDP-FedAvg and the ideal-case of FedAvg without DP (Column 2).

Table 10: Evaluating the impact of saving-to-spending transition rounds of different privacy groups, $(T_{\text{group},1}, T_{\text{group},2}, T_{\text{group},3})$, on our time-adaptive DP-FL scheme in comparison with the baseline. We set $(\epsilon_{\text{group},1}, \epsilon_{\text{group},2}, \epsilon_{\text{group},3}) = (10, 20, 30)$, $T = 25$, $L = 30$, $N = 100$, $(q_{\text{group},1}, q_{\text{group},2}, q_{\text{group},3}) = (0.3, 0.5, 0.7)$, $q = 0.9$, $c = 250$, $\lambda = 0.001$, and $B = 125$.

| DATASET | FedAvg non-DP | Ours $(7, 7, 7)$ | Ours $(7, 13, 19)$ | Ours $(19, 13, 7)$ | Ours $(19, 19, 19)$ | IDP-FedAvg |
|---------|---------------|------------------|--------------------|--------------------|---------------------|------------|
| MNIST | 90.23 | **74.38** | **74.69** | **72.24** | **73.88** | 64.53 |
| FMNIST | 72.95 | **66.72** | **65.29** | **65.34** | **67.5** | 62.57 |

**Experiments on The CIFAR10 Dataset.** We run experiments on the CIFAR10 dataset. The final-round test accuracies of our time-adaptive DP-FL framework in comparison with the FedAvg (non-DP) and IDP-FedAvg baselines are presented in Table 11. The results suggest that our proposed approach surpasses IDP-FedAvg, by lowering the gap to the ideal case of FedAvg by about 9%. We note that the test accuracies reported for all schemes in this table are relatively lower than those we reported earlier in this paper for the MNIST, FMNIST, and Adult Income datasets. We hypothesize that this happens due to the increased complexity of the CIFAR10 dataset, particularly when distributed in a non-iid manner in an FL setting with $N = 100$ clients.

Table 11: Benchmarking our time-adaptive DP-FL framework against the baselines using the CIFAR10 dataset. We set $(\epsilon_{\text{group},1}, \epsilon_{\text{group},2}, \epsilon_{\text{group},3}) = (100, 50, 25)$, $T = 50$, $L = 30$, $N = 100$, $(q_{\text{group},1}, q_{\text{group},2}, q_{\text{group},3}) = (0.5, 0.5, 0.5)$, $q = 0.9$, $c = 250$, $\lambda = 0.001$, and $B = 125$.

| DATASET | FedAvg | IDP-FedAvg | Ours |
|---------|--------|------------|------|
| CIFAR10 | 44.42 | 34.97 | 35.41 |

