# OpenReview forum: "Differentially Private Federated Learning with Time-Adaptive Privacy Spending"
_ICLR.cc/2025/Conference — ICLR 2025 Poster_

### Official Review · Reviewer_UXeM · 2024-11-03

**Soundness:** 3
**Presentation:** 3
**Contribution:** 3
**Rating:** 8
**Confidence:** 3

**Summary:**

This paper proposes a differentially private federated learning algorithm with flexible noise injected over the training time. They do so by manipulating the client sampling rate: smaller rate for the first course of the training (saving phase) and larger rate for the later part of the training (spending phase). The author intensively analyzes the privacy accounting for both phases and shows how to select the optimal client sampling rate.

**Strengths:**

1. The idea of cutting down the privacy budget spent in the early training phase is fascinating and quite new.
2. Extensive theoretical analysis of the proposed method is given.
3. Experimental results show a moderate amount of privacy budget saved throughout the training, supporting the proposed mechanism.

**Weaknesses:**

Limited empirical comparison with other adaptive differential privacy methods.

**Questions:**

1. When do clients decide to switch from privacy saving to spending phase?
2. Is the data sensitivity taken into account when computing the differential privacy noise scale?
3. Would model heterogeneity be a challenge with the proposed algorithm?

---

> ### Author Response · Authors · 2024-11-24
> **Rebuttal**
>
> >**Limited empirical comparison with other adaptive differential privacy methods.**
>
> We implemented adaptive clipping [Ref 1] as a baseline and added the results to Table 7 in Appendix A.8. In the revised paper, we also provided relevant details on this baseline in Sections 2 and 5, and Appendices A.1 and A.8. The summary of results is provided below. We performed adaptive clipping in two versions: 1) with a fair parameter alignment to our scheme where we align parameter server-side learning rate $\lambda_{\text{s}}=1$ and the server-side momentum $\beta_{\text{s}}=0$, and 2) with optimal parameters tuned for best adaptive clipping results. As observed from the following results, even with the optimal parameter choices, *adaptive clipping underperforms our scheme*.
>
> | Dataset | Uniform privacy budgets | Adaptive clipping with fair parameters | Adaptive clipping with optimal parameters | Ours |
>
> | FMNIST | (10,10,10) | 60.23 | 67.64 | 67.9 |
>
> | FMNIST | (5,5,5) | 52.39 | 52.39 | 60.79 |
>
> | MNIST | (10,10,10) | 65.59 | 78.04 | 80.2 |
>
> | MNIST | (5,5,5) | 55.48| 55.48 | 69.07 |
>
> It is also important to note that, while targeting the same goal of improving the privacy-utility tradeoff as ours, adaptive clipping maintains a fixed privacy spending over time. Instead, adaptive clipping adjusts clipping norms non-uniformly over time. Note that this idea is orthogonal to our approach and could be combined with ours for potential performance gain. However, as mentioned in Sec. 2, such integration requires careful privacy analysis, which is beyond the scope of this paper and left for future work.
>
> **References**
>
> [Ref 1] Galen Andrew, Om Thakkar, Brendan McMahan, and Swaroop Ramaswamy. Differentially private learning with adaptive clipping. Advances in Neural Information Processing Systems, 34:17455– 17466, 2021.
>
> >**When do clients decide to switch from privacy saving to spending phase?**
>
> Throughout our experimental results in Sec. 5, and Appendix A.8 (except for Table 10), we fixed transitioning from saving to spending midway through training (i.e., $T_n=T/2$) and observed consistently this yields good results in comparison with baselines.
>
> As discussed in the “Privacy Hyperparameters” paragraph in Sec. 3, intuitively, if $T_n$ is aligned with the client $n$ transitioning from coarse-grained training in early rounds to fine-grained feature training in later rounds, the client's saved budget during early rounds enables the client to spend more in later rounds when additional accuracy is beneficial in learning more fine-grained features. However, the exact timing of the transition from coarse-grained to fine-grained learning is generally unknown. As noted in Sec. 6, hyperparameter tuning to estimate this transition could result in additional privacy loss, which we leave as a potential direction for future work.
>
> We added new evaluations for different choices of $T_n$. This is detailed in the newly added Table 10 in Appendix A.8, and is summarized below. These results also confirm the robustness of our method to the client's choice of transition rounds—showing only a marginal variation in accuracy across different choices of $T_n$ while consistently achieving significant improvement compared to the privacy-constrained baseline.
>
> For different choices of the switching times $T_{\text{group},1}, T_{\text{group},2}, T_{\text{group},3}$, and assuming the total number of rounds equals $T=25$ :
>
> |Dataset | FedAvg | Our scheme (07, 7, 7) | Our scheme (7, 13, 19) | Our scheme (19, 13, 7) |  Our scheme (19, 19, 19) | IDP-FedAvg |
>
> |MNIST  | 90.23 | 74.38  | 74.69 | 72.24 | 73.88 | 64.53 |
>
> |FMNIST| 72.95 | 66.72 | 65.29 | 65.34 | 67.5 | 62.57|
>
> >**Is the data sensitivity taken into account when computing the differential privacy noise scale?**
>
> The answer is yes. As noted in Sec. 3.1, we use $c$ to denote the global clipping norm and $c_n^t$ to denote the clipping norm of client $n$ at round $t$. As specified in line 12 of Algorithm 1 in Sec. 3, we calculate $c_n^t$ as a function of $c$ and local and global noise multipliers (noise scale). Once we approach the noise addition phase of the DP mechanism, as in line 8 in Algorithm 2 in Sec. 3.2 and line 10 in function $\texttt{ClientUpdate}$ in Algorithm 2, we produce noise from Gaussian distribution with variance $(c\times \sigma^t)^2/N$ on the server side and $(c_n^t\times \sigma_n^t)^2/N$ on the client side. By clipping the model updates to norm $c_n^t$ and calibrating the noise accordingly, we take data sensitivity into account.
>
> >**Would model heterogeneity be a challenge with the proposed algorithm?**
>
> Model heterogeneity, to our understanding, would mean that every client can have a different architecture of model. This setup is not supported by gradient-based FL where one shared model is initialized for the entire collaboration. Every client hence updates the same model of the same architecture. This is not a limitation of our method, but the general setup in FL.

---

> > ### Comment · Reviewer_UXeM · 2024-11-26
> >
> > Thank you for the author's response and clarification. I will maintain my score of the paper.

---

> > > ### Author Response · Authors · 2024-11-30
> > > **We thank the reviewer for engaging in the rebuttal and their positive assessment of the paper.**
> > >
> > > We thank the reviewer for engaging in the rebuttal and their positive assessment of the paper.

---

### Official Review · Reviewer_iRT5 · 2024-11-04

**Soundness:** 3
**Presentation:** 3
**Contribution:** 3
**Rating:** 6
**Confidence:** 4

**Summary:**

This paper addresses the problem of heterogeneous client-level DPFL with non-uniform privacy expending across time (and clients). The reasoning behind the idea is that, as claimed, saving privacy budgets in early rounds lets clients with stricter privacy requirements (with smaller clipping threshold or larger noise scales) spend privacy unevenly across rounds compared to clients with more relaxed privacy requirements. This enables them to spend more privacy in later rounds when more fine-grained features are learned.

**Strengths:**

* The problem of time-adaptive privacy spending is interesting.
* The existing related works are mentioned and discussed clearly.

**Weaknesses:**

There are multiple weaknesses in the  work as mentioned below.

* The current writing of the draft is hard to understand, and requires one to read it multiple times to get a clear understanding of the work.
* There are some big assumptions that make one to hesitate about the applicability of the work.
* Some of the theoretical analysis seem to be incorrect (see the questions below).
* The existing close algorithms (e.g., adaptive clipping threshold) are not evaluated.
* The existing datasets in the experimental results are pretty simple (MNIST and FMNIST). Running experiments with more complex datasets, e.g., CIFAR10/100, will improve the work.

**Questions:**

I ask the following questions based on their order in the draft.

* In line 169, the authors claim that it is unfair to compare with the adaptive clipping methods. However, these methods are proposed for reducing the noise level as the training continues, and are directly related to the goal of this paper, even more than IDPSGD-based method (I-DPFedAvg). As such, comparison to these baselines is crucial. Also, they have proposed some methods for reducing the extra privacy leakage for tuning clipping thresholds adaptively. So there is no barrier for comparing the proposed method with them.

* The privacy analysis done in alg.1 is performed **before** training starts. It gets $c$, $q$, $\{q_n\}$, $\{M_n^t\}$ and $T$ all as hyperparameters, and as output, returns $\{q^t\}$, $\{q_n^t\}$, and $\{c_n^t\}$. First, having this many hyperparameters before training is unrealistic and makes the applicability of the proposed method limited indeed. Second, I think the privacy analysis in alg1 is incorrect. The reason is that a client can compute the amount of privacy that it has spent **only after each round that it really participates** in the FL. The value of sampling rate $q_n^t$ implicitly determines the number of times the client $n$ participates during the total $T$ rounds. It is impossible to account for the amount of privacy a client spends before even starting training. What if simply out of randomness, the client $n$ turns out to participate in all rounds $T$ (as if its sampling rate $q_n^t$ was 1 for all $t$). How do you explain about this scenario? How can you claim that you can always run alg.1 before even starting the training, and return the set of $q_n^t$ based on its result?


* The threat model considered is based on having "honest-but-curious" clients and "honest" server. How does the analysis in your paper and the proposed method work if you consider an "honest-but-curious" server? For instance the very recent work [1] below also aims at improving utility in the system by a noise-aware aggregation (it should be added to your related works). Does your algorithm work in this threat model too?

"Noise-Aware Algorithm for Heterogeneous Differentially Private Federated Learning", ICML 2024 (https://arxiv.org/pdf/2406.03519v3)

* How do you determine the set $\{q_n^t\}$ and $\{M_n^t\}$ in your experiments? Especially, where is the latter coming from?

Some minor comments:
* line 119 and 122: $(\epsilon, \delta)$-DP is the common notation
* line 137: provide a brief introduction to SGM
* the order of lines 14 and 15 in alg1 should be exchanged.
* line 6 in alg2 needs "5"inputs (one input is missing)

---

> ### Author Response · Authors · 2024-11-24
> **Rebuttal - Part 1**
>
> >**There are multiple weaknesses in the work as mentioned below. The current writing of the draft is hard to understand, and requires one to read it multiple times to get a clear understanding of the work**
>
> Our response: Based on the reviewer’s suggestion, we worked on making our paper more accessible. Therefore, we moved the intuitive discussion on our framework design and parameter choices to earlier in Sec. 3, in the “Privacy Hyperparameters” paragraph. This replaces its previous placement in the subsections and appendix. Additionally, we added new Tables 1 and 3 with notations. If they could detail, we will be more than happy to further adjust in the revised version.
>
>
> >**some big assumptions make one to hesitate about the applicability of the work**
>
> Our response: We assume that the big assumptions that the reviewer refers to are regarding the selection of the privacy parameters and the privacy analysis before the beginning of the training, as pointed out in their questions. We provide a detailed answer to the questions below but are happy to provide further clarification if the reviewer can specify which assumptions they refer to.
> >**Some of the theoretical analysis seem to be incorrect (see the question below). Second, I think the privacy analysis in alg1 is incorrect. The reason is that a client can compute the amount of privacy that it has spent only after each round that it really participates in the FL. The value of sampling rate q_n^t  implicitly determines the number of times the client n  participates during the total T  rounds. It is impossible to account for the amount of privacy a client spends before even starting training. What if simply out of randomness, the client n turns out to participate in all rounds T  (as if its sampling rate q_n^t was 1 for all t). How do you explain about this scenario? How can you claim that you can always run alg.1 before even starting the training, and return the set of q_n^t  based on its result?**
>
> We are happy to clarify the misunderstanding: We operate in distributed DP (DDP) with a trusted server. In this model, all clients share their noisy updates with the server.  The server aggregates the updates that are selected through sampling and adds additional noise for each of the discarded ones. From the noisy aggregated updates, it is not possible to tell which client contributed to the aggregated update. Given that we assume all exchanges between server and clients are encrypted, and that the server is trusted, to the outside world, including to the clients themselves, the effect is the same as secure aggregation. In other words, not even the clients would know if their update was in the final update of a given step.
> In the revised manuscript, we make this explicit by altering Line 5 in our Algorithm 2 such that every client always sends the update (deniability), and the server then does the selection as $C^t \leftarrow$ Sample clients with probability $\{q_n^t\}$, where $n\in [N]$.
> Our privacy accounting is entirely in line with standard DP and IDP where data points receive their privacy levels *on expectation*. In practice, some might be sampled more often than others, but the results are indistinguishable.

---

> ### Author Response · Authors · 2024-11-24
> **Rebuttal - Part 2**
>
> >**The existing close algorithms (e.g., adaptive clipping threshold) are not evaluated).**
>
> We performed the suggested additional evaluation. We implemented adaptive clipping [Ref 1], with results being summarized below. In the revised paper, we added these results to Table 7 in Appendix A.8 and provided relevant details in Sections 2 and 5, and Appendices A.1 and A.8. We performed adaptive clipping in two versions: 1) with a fair parameter alignment to our scheme where we align parameter server-side learning rate $\lambda_{\text{s}}=1$ and the server-side momentum $\beta_{\text{s}}=0$, and 2) with optimal parameters tuned for best adaptive clipping results. As observed from the following results, even with the optimal parameter choices, *adaptive clipping underperforms our scheme*.
>
> [Ref 1] Galen Andrew, Om Thakkar, Brendan McMahan, and Swaroop Ramaswamy. Differentially private learning with adaptive clipping. Advances in Neural Information Processing Systems, 34:17455– 17466, 2021.
>
> | Dataset | Uniform privacy budgets | Adaptive clipping with fair parameters | Adaptive clipping with optimal parameters | Ours |
>
> | FMNIST | (10,10,10) | 60.23 | 67.64 | 67.9 |
>
> | FMNIST | (5,5,5) | 52.39 | 52.39 | 60.79 |
>
> | MNIST | (10,10,10) | 56.59 | 78.04 | 80.2 |
>
> | MNIST | (5,5,5) | 55.48| 55.48 | 69.07 |
>
> It is also important to note that, while targeting the same goal of improving the privacy-utility tradeoff as ours, adaptive clipping maintains a fixed privacy spending over time. Instead, adaptive clipping adjusts clipping norms non-uniformly over time. Note that this idea is orthogonal to our approach and could be combined with ours for potential performance gain. However, as mentioned in Sec. 2, such integration requires careful privacy analysis, which is beyond the scope of this paper and left for future work.
>
>
> >**The existing datasets in the experimental results are pretty simple (MNIST and FMNIST). Running experiments with more complex datasets, e.g., CIFAR10/100, will improve the work.**
>
> We performed the evaluation on the CIFAR10 dataset. The results are summarized below. In the revised paper, we added these results to Table 11 in Appendix A.8 and provided relevant details in Section 5 and Appendix 8.
>
> | Dataset | IDP-FedAvg | our scheme (non-uniform) | FedAvg (w/o DP) |
>
> | CIFAR10 | 34.5 | 35.41 | 44.42|
>
> The results suggest that our proposed approach surpasses IDP-FedAvg, by lowering the gap to the ideal case of FedAvg by about $9$%. We note that the test accuracies reported for all schemes in this table are relatively lower than those we reported earlier in this paper for the MNIST, FMNIST, and Adult Income datasets. We hypothesize that this happens due to the increased complexity of the CIFAR10 dataset, particularly when distributed in a non-iid manner in an FL setting with $N=100$ clients.

---

> ### Author Response · Authors · 2024-11-24
> **Rebuttal - Part 3**
>
> >**The privacy analysis done in alg.1 is performed before training starts. It gets $c$, $q$, $q_n$, $M_n^t$  and $T$  all as hyperparameters, and as output, returns $q^t$ , $q_n^t$, and $c_n^t$. First, having this many hyperparameters before training is unrealistic and makes the applicability of the proposed method limited indeed.**
>
> We note that obtaining the privacy parameters (which corresponds to determining the privacy analysis) before training is a standard practice both in standard DP and in IDP. For example, the Opacus library that implements DP calls `get_noise_multiplier` [1] before starting training to determine how much noise needs to be added to every training step to obtain the final epsilon. Also, the sampling rate is determined before training based on the dataset size [2].
> On the contrary to limiting the applicability of the method, we argue that determining the hyperparameters once before training increases the applicability of the method since it requires only *one* additional communication step between the server and clients to communicate the determined ones. The clients can then independently operate with the received parameters.
>
> [1] https://github.com/pytorch/opacus/blob/61ae0ea4fb37a835e93040b5de19e8dfcd465a07/opacus/accountants/utils.py#L23
>
> [2] https://github.com/pytorch/opacus/blob/61ae0ea4fb37a835e93040b5de19e8dfcd465a07/opacus/privacy_engine.py#L381
>
> >**The threat model considered is based on having "honest-but-curious" clients and "honest" server. How does the analysis in your paper and the proposed method work if you consider an "honest-but-curious" server?**
>
>  In response to the reviewer's comment, we note that our privacy analysis assumes a trusted server. However, by using a Distributed DP (DDP) implementation, where clients clip their updates and add partial noise, our approach offers privacy protection against the server, albeit at a lower level. As discussed in Sec. 3, this is because, once the server aggregates the perturbed updates, the total noise increases, enhancing privacy protection against other clients but less against the server. Due to the reduced noise per client update ($1/N$) and the server's full knowledge of sampling sets, the RDP privacy bounds against the server are weakened (increases), scaling by a factor of (number of clients)/sampling rate^2. Adapting our approach to the suggested paper’s framework would require shifting all server-side operations, including sampling, aggregation, and noise addition, to a secure multiparty computation scheme among the clients, with the updates aggregated using secure aggregation protocols. We have included this potential extension as future work in the “Threat Space” paragraph in Sec. 3.
>
> >**For instance the very recent work [1] below also aims at improving utility in the system by a noise-aware aggregation (it should be added to your related works).**
>
> We thank the reviewer for pointing out this reference. We added it to our related work section, Sec. 2, in the “personalized DP” paragraph.
>
> >**How do you determine the set $q_n^t$  and $M_n^t$  in your experiments? Especially, where is the latter coming from?**
>
> In response to this comment, we have revised Sec. 3.1 to further clarify the determination of $q_n^t$ and $M_n^t$. First we have clarified that $M_n^t$ is a binary variable that is set to 0 if round $t$ is a saving round for client $n$, and 1 otherwise. I.e., $M_n^t = 0$ if $t< T_n$ and $M_n^t = 1$ if $t\geq T_n$. As noted in the “Privacy Hyperparameter” paragraph in Sec. 3, $T_n$ denotes the first round when client $n$ is in the spend mode.
>
>  Regarding $q_n^t$, we have clarified that is chosen as either $q$ or $q_n$, depending on whether client $n$ is in spend or saving mode. As noted in the “Privacy Hyperparameter” paragraph in Sec. 3, $q_n$ and $q$ denote the sampling rates during saving and spending modes, respectively.
>
> >**Some minor comments:**
>
> We thank the reviewer for bringing these to our attention. In the revised manuscript, we corrected all points they raised and other typos that we found.

---

> > ### Comment · Reviewer_iRT5 · 2024-11-26
> > **Feedback to the authors**
> >
> > Thanks for the detailed responses. There have been some misunderstandings and concerns on my side which are now resolved. I have raised my score accordingly.

---

> > > ### Author Response · Authors · 2024-11-26
> > > **We thank the reviewer for engaging in the rebuttal and raising their score.**
> > >
> > > We thank the reviewer for engaging in the rebuttal and raising their score.

---

### Official Review · Reviewer_19qN · 2024-11-04

**Soundness:** 3
**Presentation:** 3
**Contribution:** 3
**Rating:** 8
**Confidence:** 3

**Summary:**

The paper studies federated learning with distributed differential privacy, where each client adds noise to its model update/gradient before sending to the server, and a trusted server aggregates the noisy updates for an overall $(\epsilon,\delta)$ aggregated model. The authors point out that the early rounds are less sensitive to noise while later rounds are heavily affected by the introduction of noise. Hence, the paper proposes that privacy budget can be saved in early rounds so that the later rounds can spend more privacy budget as they need it more. To achieve this, the paper proposes to use a lower sub-sampling rate in earlier rounds and normal sampling rate in the later rounds/spending rounds. The paper proves that given a set of sampling rates, one can optimize the assignment of these sampling rates to clients (based on their personalized privacy budget) to minimize an upper bound of the expected clipping bias. Empirical evaluation shows that spend-as-you-go achieves higher performance under the same privacy budgets.

**Strengths:**

* Improving the performance of DP-FL is an important and interesting problem
* The adaptive spending scheme is intuitive and well motivated
* Theoretical analysis provides insights on the algorithm and bounds the expected clipping bias
* The paper is well written and easy to read

**Weaknesses:**

* In section 3, the algorithm for the spend-as-you-go budgeting scheme and the FL training loop are described in great details. However, the motivation of each design choice is not clearly explained. I suggest the authors to edit the section to include explain the rationale behind each design choice that defers from the conventional FL algorithm.
* The configuration of the hyperparamters of the proposed method are not explained. For example, how are the set of lower sampling rates determined? Why do we choose the values presented in the Appendix? Also, how is the switching round $T_n$ chosen?
* The paper could benefit from additional experiments.
  * I would like to see the empirical evaluation on the effects of the set of sampling rates and the switching time.
  * I suggest the authors to also include the non-private baseline in the figures for reference.
  * If I understand correctly, the proposed adaptive spending scheme also works for non-personalized FL (all clients share the same budget) as well as non-FL (i.e., DP-SGD). Will the spending scheme also bring performance improvements? It would be interesting to see how the proposed scheme would perform in these settings.

**Questions:**

I raise a few questions in the earlier section of weaknesses when I list my concerns. I would appreciate it if the authors could address the questions listed above.

---

> ### Author Response · Authors · 2024-11-24
> **Rebuttal - Part 1**
>
> >**In section 3, the algorithm for the spend-as-you-go budgeting scheme and the FL training loop are described in great details. However, the motivation of each design choice is not clearly explained. I suggest the authors to edit the section to include explain the rationale behind each design choice that defers from the conventional FL algorithm.**
>
> We moved the intuitive discussion on our framework design and parameter choices to earlier in Sec. 3, in the “Privacy Hyperparameters” paragraph. This replaces its previous placement in the subsections and appendix.
>
> >**The configuration of the hyperparamters of the proposed method are not explained. For example, how are the set of lower sampling rates determined?**
>
> We have clarified that the lower sampling rates—or what is called saving-based sampling rates $q_1,\ldots,q_N$ in the paper—must satisfy $q_n \in [0,q]$. As discussed in the “Privacy Hyperparameters” paragraph in Sec. 3, it is intuitive that if $q_n<q$ then client $n$ is less likely to be sampled during saving versus spending rounds. According to privacy amplification by sub-sampling (Beimel et al, 2014), and as detailed in Sec. 4.1, a fraction of the clients’ privacy budget will then be saved for later rounds.
>
> Also, as shown in the theoretical analysis in Sec. 4.2, the rates $q_n$ should be ordered consistently with privacy budgets $\epsilon_n$. As discussed in Sec. 1, intuitively, this allows less-privacy-sensitive clients, who have often sufficient budgets, to contribute to the learning both of coarse-grained features in early rounds and of fine-grained features in later rounds. On the other hand, more privacy-constrained clients can preserve their budgets and helpfully contribute more to the learning of fine-grained features. In our experiments, we select a set of $q_n$ that satisfy these constraints. However, as noted in Sec. 6, we reiterate that tuning these parameters can incur a privacy loss. Therefore, we run experiments with only a few different choices of $(q_{\text{group},1},q_{\text{group},2},q_{\text{group},3})$ from the set $\{(0.3,0.5,0.7), (0.5,0.6,0.7)\}$ for MNIST/FMNIST/CIFAR10 and from $\{(0.3,0.5,0.7), (0.5,0.6,0.7), (0.6,0.7,0.8)\}$ for Adult-income, and report the best results.
>
> It is important to note that we observed our method is robust against the clients' choice of saving-based sampling rates—while a different improvement was achieved with a different choice of $q_n$, all tested choices of $q_n$ consistently achieved performance better than the other privacy-constrained baselines. This is detailed in the newly added Table 9 in App. A.8, where we added new evaluations for different choices of $q_n$. These results are provided in response to the reviewer in a separate comment.
>
> >**Why do we choose the values presented in the Appendix?**
>
> We ran our method under multiple parameter selections and present the final results in the main paper. Ablations on other parameter choices can be found in Appendix A.7 and Appendix A.8. If the reviewer has specific questions on some specific parameter choices, we are happy to further clarify these details.
>
> >**Also, how is the switching round T_n chosen?**
>
> Throughout our experimental results in Sec. 5, and Appendix A.8 (except for Table 10), we fixed transitioning from saving to spending midway through training (i.e., $T_n=T/2$) and observed consistently this yields good results in comparison with baselines. As discussed in the “Privacy Hyperparameters” paragraph in Sec. 3, intuitively, if $T_n$ is aligned with the client $n$ transitioning from coarse-grained training in early rounds to fine-grained feature training in later rounds, the client's saved budget during early rounds enables the client to spend more in later rounds when additional accuracy is beneficial in learning more fine-grained features. However, the exact timing of the transition from coarse-grained to fine-grained learning is generally unknown. As noted in Sec. 6, hyperparameter tuning to estimate this transition could result in additional privacy loss, which we leave as a potential direction for future work.
>
> As suggested by the reviewer in another comment, we added new evaluations for different choices of $T_n$. This is detailed in the newly added Table 10 in Appendix A.8. These results also confirm the robustness of our method to the client's choice of transition rounds—showing only a marginal variation in accuracy across different choices of $T_n$ while consistently achieving significant improvement compared to the privacy-constrained baseline.

---

> ### Author Response · Authors · 2024-11-24
> **Rebuttal - Part 2**
>
> >**The paper could benefit from additional experiments.**
>
> Extended experimental results, including benchmarks on the CIFAR10 dataset, comparisons with adaptive clipping, and evaluations across parameters, are added to Appendix A.8. While results for evaluations across parameters are provided in response to the other comments by the reviewer, below we summarize our results on the CIFAR10 dataset and comparisons with adaptive clipping.
>
> Results for CIFAR10 are summarized below. In the revised paper, we added these results to Table 11 in Appendix A.8 and provided relevant details in Section 5 and Appendix 8.
>
> | Dataset | IDP-FedAvg | our scheme (non-uniform) | FedAvg (w/o DP) |
>
> | CIFAR10 | 34.5 | 35.41 | 44.42|
>
> Results for the adaptive clipping [Ref 1] baseline are summarized below. In the revised paper, we added these results to Table 7 in Appendix A.8 and provided relevant details in Sections 2 and 5, and Appendices A.1 and A.8.
> We performed adaptive clipping in two versions: 1) with a fair parameter alignment to our scheme where we align parameter server-side learning rate $\lambda_{\text{s}}=1$ and the server-side momentum $\beta_{\text{s}}=0$, and 2) with optimal parameters tuned for best adaptive clipping results. As observed from the following results, even with the optimal parameter choices, *adaptive clipping underperforms our scheme*.
>
> [Ref 1] Galen Andrew, Om Thakkar, Brendan McMahan, and Swaroop Ramaswamy. Differentially private learning with adaptive clipping. Advances in Neural Information Processing Systems, 34:17455– 17466, 2021.
>
> | Dataset | Uniform privacy budgets | Adaptive clipping with fair parameters | Adaptive clipping with optimal parameters | Ours |
>
> | FMNIST | (10,10,10) | 60.23 | 67.64 | 67.9 |
>
> | FMNIST | (5,5,5) | 52.39 | 52.39 | 60.79 |
>
> | MNIST | (10,10,10) | 56.59 | 78.04 | 80.2 |
>
> | MNIST | (5,5,5) | 55.48| 55.48 | 69.07 |
>
>
> >**I would like to see the empirical evaluation on the effects of the set of sampling rates and the switching time**
>
> We performed the experiment as suggested by the reviewer and added results to Appendix A.8, Tables 9 and 10. Below we provide a summary of these results.
>
> For different choices of sampling rates $q_{\text{group},1}, q_{\text{group},2}, q_{\text{group},3}$:
>
> |Dataset | FedAvg | Our scheme (0.5, 0.6, 0.7) | Our scheme (0.3, 0.5, 0.7) | Our scheme (0.6, 0.6, 0.6) | IDP-FedAvg |
>
> |MNIST  | 90.23 | 72.72  | 77.39 | 71.6 | 64.53 |
>
> |FMNIST| 72.95 | 70.57 | 66.55 | 67.75 | 62.57 |
>
> As shown in the above table, our scheme, which uses lower sampling rates during saving outperforms the IDP-FedAvg baseline that uses a uniform sampling rate over time. This table also shows that our method is relatively robust against the clients' choice of saving-based sampling rates, consistently achieving performance between that of IDP-FedAvg and the ideal-case of FedAvg without DP.
>
> For different choices of the switching times $T_{\text{group},1}, T_{\text{group},2}, T_{\text{group},3}$, and assuming the total number of rounds equals $T=25$ :
>
> |Dataset | FedAvg | Our scheme (7, 7, 7) | Our scheme (7, 13, 19) | Our scheme (19, 13, 7) | Our scheme (19, 19, 19) | IDP-FedAvg |
>
> |MNIST  | 90.23 | 74.38  | 74.69 | 72.24 | 73.88 | 64.53 |
>
> |FMNIST| 72.95 | 66.72 | 65.29 | 65.34 | 67.5 | 62.57|
>
> As shown in the above table, our scheme, which transitions from saving to spend mode sometime between the first and final round outperforms the IDP-FedAvg baseline which can be viewed as a special case of ours with transition rounds set to $(1,1,1)$. This table shows the robustness of our method to the client's choice of transition rounds, showing relatively low variation in accuracy across different choices while consistently achieving performance between that of IDP-FedAvg and the ideal case of FedAvg without DP.

---

> ### Author Response · Authors · 2024-11-24
> **Rebuttal - Part 3**
>
> >**I suggest the authors to also include the non-private baseline in the figures for reference.**
>
> Following the reviewer’s suggestion, we added results for the FedAvg baseline (non-DP scenario) to Table 2 in Sec. 5, and Tables 5, 9, 10, and 11 in Appendix A.8. We also provided relevant details in Sections 2 and 5, and Appendices A.1 and A.8. These results are summarized below. As expected and shown by these results, FedAvg provides an upper bound on the utility without any privacy constraints. However, as it is shown in the aforementioned tables in the paper, our scheme among other private baselines comes closest to the ideal-case performance of FedAvg.
>
> | Dataset | (No. clients, No. global rounds, No. local iterations) | FedAvg (w/o DP) | Ours with privacy budgets (10,20,30) for MNIST/FMNIST/Adult Income and (100,50, 25) for CIFAR10 |
>
> | FMNIST | (100, 25, ,30) | 72.95  | 66.55 |
>
> | FMNIST | (100, 50, ,30) | 76.0 | 75.51  |
>
> | FMNIST | (100, 100, ,30) | 80.14  | 75.63 |
>
> | MNIST |  (100, 25, 30) | 90.23  | 74.69 |
>
> | MNIST |  (100, 50, 30) | 93.42.23  | 90.78 |
>
> | MNIST |  (100, 100, 30) | 95.91  | 90.15 |
>
> | Adult Income | (80, 25, 5) | 78.93 | 77.53 |
>
> | CIFAR 10 | (100, 50, 30)  | 44.42  | 35.41 |
>
> >**If I understand correctly, the proposed adaptive spending scheme also works for non-personalized FL (all clients share the same budget) as well as non-FL (i.e., DP-SGD). Will the spending scheme also bring performance improvements? It would be interesting to see how the proposed scheme would perform in these settings.**
>
> We clarified that we operate in a federated setting and central learning (non-FL) is not comparable, but most likely it can benefit from adaptive time spending. Thus, it would be an interesting future work to extend our idea to centralized setting.

---

> > ### Comment · Reviewer_19qN · 2024-11-25
> >
> > Thanks the authors to address my concerns. The paper could clearly benefit from the additional experimental results provided in the authors' response. I've raised my score.

---

> > > ### Author Response · Authors · 2024-11-26
> > > **We thank the reviewer for engaging in the rebuttal and raising their score.**
> > >
> > > We thank the reviewer for engaging in the rebuttal and raising their score.

---

### Official Review · Reviewer_s8p8 · 2024-11-04

**Soundness:** 3
**Presentation:** 2
**Contribution:** 2
**Rating:** 6
**Confidence:** 3

**Summary:**

The paper presents a novel approach for DP FL wherein instead of making the clients spend the privacy budget uniformly over all communication rounds, as in traditional DP FL approaches, the clients allocate their privacy budgets non-uniformly over time. Specifically, the method suggests that clients save their privacy budgets in early communication rounds since coarse-grained features are learnt in early rounds and spend more in later rounds where more fine grained features are learned.

**Strengths:**

1. The approach is novel and intuitive.
2. The privacy analysis of the algorithm is provided.

**Weaknesses:**

1. Insufficient experimental evaluation - Only two datasets and baselines are considered.

**Questions:**

1. What is the $\delta$ for experiments in Table 1 and Figure 2? Can you elaborate on why such an high $\epsilon$ in 10, 20, 30 is acceptable? How many number of communication rounds are considered?

Minor:
1. Clearly defining notations, and relationships between different notations would make paper easier to understand, like DP to RDP mapping.

---

> ### Author Response · Authors · 2024-11-24
> **Rebuttal - part 1**
>
> >**Insufficient experimental evaluation - Only two datasets and baselines are considered.**
>
> We would like to note that our initial submission considered *three* datasets, MNIST, FMNIST, and Adults Income. Over the course of the revision, we have added experiments on a new dataset and two additional baselines, as discussed below.
>
> Results on the CIFAR10 dataset are summarized below. In the revised paper, we added these results to Table 11 in Appendix A.8 and provided relevant details in Section 5 and Appendix 8.
>
> | Dataset | IDP-FedAvg | our scheme (non-uniform) | FedAvg (w/o DP) |
>
> | CIFAR10 | 34.5 | 35.41 | 44.42|
>
> We also conducted experiments for more baselines. Results for the adaptive clipping [Ref 1] baseline are summarized below. In the revised paper, we added these results to Table 7 in Appendix A.8 and provided relevant details in Sections 2 and 5, and Appendices A.1 and A.8. We performed adaptive clipping in two versions: 1) with a fair parameter alignment to our scheme where we align parameter server-side learning rate $\lambda_{\text{s}}=1$ and the server-side momentum $\beta_{\text{s}}=0$, and 2) with optimal parameters tuned for best adaptive clipping results. As observed from the following results, even with the optimal parameter choices, *adaptive clipping underperforms our scheme*.
>
> [Ref 1] Galen Andrew, Om Thakkar, Brendan McMahan, and Swaroop Ramaswamy. Differentially private learning with adaptive clipping. Advances in Neural Information Processing Systems, 34:17455– 17466, 2021.
>
> | Dataset | Uniform privacy budgets | Adaptive clipping with fair parameters | Adaptive clipping with optimal parameters | Ours |
>
> | FMNIST | (10,10,10) | 60.23 | 67.64 | 67.9 |
>
> | FMNIST | (5,5,5) | 52.39 | 52.39 | 60.79 |
>
> | MNIST | (10,10,10) | 56.59 | 78.04 | 80.2 |
>
> | MNIST | (5,5,5) | 55.48| 55.48 | 69.07 |
>
> We also added results for the FedAvg baseline (non-DP scenario). These are summarized below. In the revised paper, we added these results to Tables 2, 5, 9, 10, and 11 and provided relevant details in Sections 2 and 5, and Appendices A.1 and A.8. As expected, FedAvg provides an upper bound on utility that can be achieved without any privacy constraints. However, as it is shown in the above tables in the paper, our scheme comes closest to the ideal-case performance of FedAvg in comparison to the other private baselines.
>
> | Dataset | (No. clients, No. global rounds, No. local iterations) | FedAvg (w/o DP) | Ours with privacy budgets (10,20,30) for MNIST/FMNIST/Adult Income and (100,50, 25) for CIFAR10 |
>
> | FMNIST | (100, 25, ,30) | 72.95  | 66.55 |
>
> | FMNIST | (100, 50, ,30) | 76.0 | 75.51  |
>
> | FMNIST | (100, 100, ,30) | 80.14  | 75.63 |
>
> | MNIST |  (100, 25, 30) | 90.23  | 74.69 |
>
> | MNIST |  (100, 50, 30) | 93.42.23  | 90.78 |
>
> | MNIST |  (100, 100, 30) | 95.91  | 90.15 |
>
> | Adult Income | (80, 25, 5) | 78.93 | 77.53 |
>
> | CIFAR 10 | (100, 50, 30)  | 44.42  | 35.41 |
>
>
> >**What is the δ  for experiments in Table 1 and Figure 2?**
>
> This is specified in the “Privacy Settings” paragraph in Sec. 5.  For our experiments $\delta=10^{-5}$. This value of $\delta$ corresponds to the standard range of values of $\delta$ used in the literature.

---

> ### Author Response · Authors · 2024-11-24
> **Rebuttal - part 2**
>
> >**Can you elaborate on why such an high ϵ  in 10, 20, 30 is acceptable?**
>
> We observe in our experiments and in prior work [Ref1, Ref2, Ref3] that with 100 or fewer clients, very small values of $\varepsilon$ do not yield high utility. This is because of the addition of high amounts of noise vs. the small magnitude signal from the few clients.
> Some examples of highly-cited/well-known work on client-level DP-FL may help underscore this point. In [Ref1] and [Ref2], the number of clients ranges from 5 to 100. In [Ref1], under the Local Differential Privacy (LDP) model, $\epsilon$ is fixed at 8. In [Ref2], under the Distributed Differential Privacy (DDP) model, values of $\epsilon$ are evaluated from the set {6, 8, 10, 50, 60, 80, 100}. In contrast, [Ref3], which relies on the Central Differential Privacy (CDP) model and assumes a client count exceeding $10^5$, reports an $\epsilon$ values of around 1 after 10 rounds.
>
> **References**
>
> [Ref1] Geyer, R.C., Klein, T. and Nabi, M., 2017. Differentially private federated learning: A client level perspective. arXiv preprint arXiv:1712.07557.
>
> [Ref2] Wei, K., Li, J., Ding, M., Ma, C., Yang, H.H., Farokhi, F., Jin, S., Quek, T.Q. and Poor, H.V., 2020. Federated learning with differential privacy: Algorithms and performance analysis. IEEE transactions on information forensics and security, 15, pp.3454-3469.
>
> [Ref3] McMahan, H.B., Ramage, D., Talwar, K. and Zhang, L., 2017. Learning differentially private recurrent language models. arXiv preprint arXiv:1710.06963.
>
> To further address the reviewer’s comment, we performed additional experiments. These experiments show that our method continues to be effective under stronger privacy requirements. Under non-uniform privacy settings, we chose $\varepsilon$ to be  2, 5, 10 and provide results in Appendix A.8, Table 6, and Figure 5. Under uniform privacy settings, we choose $\varepsilon$ to be 5 across all clients and provide results in Appendix A.8, Table 7. As shown there, and summarized below, our scheme still outperforms the baseline under these stricter privacy requirements.
>
> For non-uniform privacy budgets:
>
> | Dataset | Privacy budgets ($\epsilon_{\text{group},1}, \epsilon_{\text{group},2}, \epsilon_{\text{group},3}$) | IDP-FedAvg | Our scheme |
>
> | FMNIST |(10,20,30) | 62.57 | 66.55 |
>
> | FMNIST |(2, 5, 10) | 60.99  | 65.75 |
>
> | MNIST |(10,20,30) | 64.53 | 77.38 |
>
> | MNIST |(2, 5, 10) | 63.35 | 66.50 |
>
> For uniform privacy budgets:
>
> | Dataset | Privacy budgets| DP-FedAvg | Adaptive Clipping | Our scheme |
>
> | FMNIST |(10, 10, 10) | 64.8| 67.64 | 67.90 |
>
> | FMNIST |(5, 5, 5) | 51.06 | 52.39 | 60.79 |
>
> | MNIST |(10, 10, 10) | 76.79 | 78.04 | 80.2 |
>
> | MNIST |(5, 5, 5) | 61.45 | 55.48 | 69.07 |
>
> >**How many number of communication rounds are considered?**
>
> For the results provided in the main paper, for the MNIST and FMNIST datasets we used 25 global rounds and 30 local iterations per round. For the Adult Income dataset, we used 25 global rounds and 5 local iterations per round. For the results provided in App. A.8, we used 50 global rounds in Figures 4 and 5, and used $T= 25$, $50$, and $100$ rounds in Table 5.
>
> In the previous version of the manuscript, we summarized some of the algorithm parameters in Appendix 7 and Sec. 5. To improve clarity, we have now revised and expanded the experimental setup discussion, modifying Table 4, and discussing setups in captions to the figures and tables in the paper.
>
> This comment also motivated us to run experiments for a larger number of global rounds, and as we see below (the newly added Table 5 in Appendix A.8), we observe that the model utility increases as we increase the number of training rounds to 50, beyond which the utility degrades or stays constant when number of training rounds equals 100. We hypothesize that this happens because, as the training rounds increase, the privacy budget is distributed among the increased number of rounds, resulting in a lesser privacy budget available for every round. Thus, more perturbation is added, as the number of training rounds increases. These results are summarized below.
>
> | Dataset || Number of Rounds (T)| FedAvg | IDP-FedAvg| Ours |
>
> | FMNIST ||25 | 72.95 | 62.57 | 66.55 |
>
> | FMNIST ||50 | 76.00 | 71.8 | 75.51 |
>
> | FMNIST ||100 | 80.14 | 71.29 | 75.63 |
>
> | MNIST||25 | 90.23 | 64.53 | 74.69 |
>
> | MNIST ||50 | 93.42 | 89.57 | 90.78 |
>
> | MNIST ||100 | 95.91 | 87.0 | 90.15 |
>
>
> >**Clearly defining notations, and relationships between different notations would make paper easier to understand, like DP to RDP mapping.**
>
> In the revised manuscript we expanded Appendix A.6. We renamed it “Extended Background” and added detailed notations and definitions of DP and RDP. We note that Lemma 3 in that section discusses how to map between DP and RDP, we refer to this Lemma in Sec. 2.

---

> ### Author Response · Authors · 2024-11-28
> **Rebuttal follow up**
>
> We appreciate the reviewer's time and efforts. We would like to learn whether we have addressed the reviewer’s concerns. We would be pleased to follow up with additional clarifications as needed.

---

> > ### Comment · Reviewer_s8p8 · 2024-12-02
> > **Response to Authors' Rebuttal**
> >
> > Thanks to the authors for the detailed response to my comments. I have increased my score in response to the authors' rebuttal.
> >
> > Thank you!

---

> > > ### Author Response · Authors · 2024-12-02
> > > **We thank the reviewer for engaging in the rebuttal and raising their score.**
> > >
> > > We thank the reviewer for engaging in the rebuttal and raising their score.

---

### Author Response · Authors · 2024-11-24
**Rebuttal**

We thank all reviewers for their helpful comments and suggestions. We also appreciate the interest they all show in our problem and the recognition of its novelty and intuitive underlying ideas (s8p8, 19Qn, UXeM). Particularly, we are glad that the reviewers recognize our work to be interesting (s8p8, 19qN, iRT5, UXem), novel (s8p8, UXeM), intuitive (s8p8, 19qN), well written (19qN), supported by both theoretical analysis (s8p8, 19qN, UXeM) as well experimental results (UXem), that we reviewed related work clearly (iRT5), and that we address an important problem (19qN). Based on the reviewers’ comments, we made the following main changes to improve our manuscript:

1. We clarified our setup to address Reviewer iRT5’s concern about determining privacy parameters before training starts. In summary, we operate in distributed DP (DDP) with a trusted server handling sampling.  This means that neither clients nor any external party knows which client updates are included in the aggregated update in any given round. This approach aligns completely with standard DP and IDP privacy accounting.  It also aligns with the way that standard software libraries, such as Opacus, are implemented.  In these implementations, all points receive their privacy budget "in expectation". While some clients may be sampled more frequently in practice, the specific sampling pattern realized remains unknown to all parties. To clarify this point in our manuscript, we make revisions in the paper.  We detail the revisions in our direct response to the reviewer.

2. To address the suggestions about extending the experimental results, in the revised manuscript we inserted additional experiments using the  (new) dataset CIFAR10 (this is detailed in our direct responses to Reviewers s8p8’s and iRT5’s comments), new baselines that including adaptive clipping (as detailed in our direct responses to Reviewers s8p8’s, iRT5’s, and UXeM’s comments) and FedAvg without DP (as detailed in our direct response to Reviewer 19qN’s comments), and across a broader range of hyperparameters (as detailed in our direct response to Reviewer 19qN’s comments). We note that results on non-personalized (uniform) privacy budgets were already included in our initial submission. We have expanded on this. Further details are provided in our response to Reviewer 19qN’s comment.

3. We addressed the reviewers’ comments on the paper’s structure regarding the presentation of hyperparameter configuration and assumptions. In the revised manuscript we moved more of the discussion from the appendix to Sec. 3, adding a new paragraph and a new table. More details are provided in our response to Reviewer 19qN’s comment.

---

### Author Response · Authors · 2024-11-28
**Rebuttal Pending Questions**

We would like to thank the reviewers for their questions and comments. The paper has definitely improved as a result. We would like to check one last time if there are any pending questions that we have not adequately addressed.

---

### Meta-Review · Area_Chair_iQ8t · 2024-12-20

**Metareview:**

The reviewers are excited about the paper, and more specifically the spend as you go privacy accounting mechanism. The reviewers also found the impact of the scheme on the clipping bias to be important. We think the authors have adequately addressed all the concerns, and updated the draft accordingly.

**Additional Comments On Reviewer Discussion:**

See above.

---

### Decision · Program_Chairs · 2025-01-22

Accept (Poster)